# On Group Relative Policy Optimization Collapse in Agent Search: The Lazy Likelihood-Displacement

Wenlong Deng [◇ * 1 2]   Yushu Li [* 1 2]   Boying Gong [* 3]   Yi Ren [4]   Christos Thrampoulidis [† 1]   Xiaoxiao Li [† 1 2]

## Abstract

Tool-integrated (TI) reinforcement learning (RL) enables large language models (LLMs) to perform multi-step reasoning by interacting with external tools such as search engines. Group Relative Policy Optimization (GRPO), exemplified by the recent Search-R1 (Jin et al., 2025), offers fast convergence and a value-free formulation that makes it appealing for this setting, yet consistently suffers from training collapse. We identify Lazy Likelihood Displacement (LLD), a systematic reduction or stagnation in the likelihood of correct response trajectories, as the core mechanism driving this failure. LLD emerges early and triggers a self-reinforcing LLD Death Spiral, where declining likelihood leads to low-confidence responses, inflating gradients and ultimately causing collapse. We empirically characterize this process across models on a search-integrated question answering task, revealing a three-phase trajectory: early stagnation, steady decay, and accelerated collapse. To address this, we propose a likelihood-preserving regularization LLDS that activates only when a response action's likelihood decreases, and regularizes only the tokens responsible. This fine-grained structure mitigates LLD with minimal interference. Our method stabilizes training, prevents gradient explosion, and yields substantial performance improvements across seven benchmarks, including relative improvements of **+45.2%** on Qwen2.5-3B and **+37.1%** on Qwen2.5-7B over vanilla GRPO training. Our results establish LLD as a previously overlooked bottleneck in GRPO-based TIRL and provide a practical path toward stable, scalable training of tool-integrated RL. Code is available at ⍟ **Code**.

[*]Equal contribution. [◇]Project Leader   [†]Corresponding authors.   [1]University of British Columbia [2]Vector Institute [3]Meta AI [4]University of Oxford; For any inquiries, please contact dwenlong@student.ubc.ca.

*Proceedings of the $43^{rd}$ International Conference on Machine Learning*, Seoul, South Korea. PMLR 306, 2026. Copyright 2026 by the author(s).

## 1. Introduction

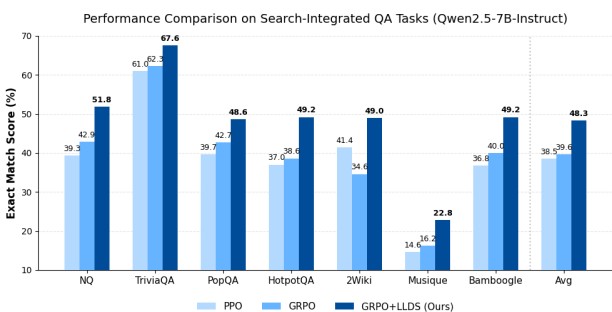

*Figure 1.* Comparative performance of LLDS and baseline methods on benchmark datasets. All baselines are built upon Qwen2.5-7B-Instruct. See Section 6.1 for details.

Large language models (LLMs) increasingly leverage external tools, such as search engines (Jin et al., 2025) and code-execution environments (Feng et al., 2025; Zeng et al., 2025), to augment their reasoning capabilities. This tool-integrated reasoning (TIR) paradigm has driven recent progress across factual question answering (QA) (Jin et al., 2025), image-based reasoning (Jiang et al., 2025), and mathematical problem solving (Feng et al., 2025; Jiang et al., 2025). By enabling models to iteratively query tools, tool calls substantially elevate reasoning quality (Qin et al., 2023). These advances naturally motivate the use of reinforcement learning (RL) to train LLMs to plan, interact with tools, and master multi-step decision making, as instantiated by TIRL frameworks such as Search-R1 (Jin et al., 2025). However, extending TIR to the RL setting leads to severe and persistent training instabilities. In particular, GRPO training in Search-R1–style pipelines frequently suffers from abrupt reward drops and catastrophic collapse (Jin et al., 2025; Sun et al., 2025). These failures are especially pronounced in multi-turn tool scenarios (Xue et al., 2025), where out-of-distribution tool feedback is incorporated into the model's context. Although prior work observed these failures, the underlying mechanisms lack understanding.

In this work, we first identify Lazy Likelihood Displacement (LLD) (Deng et al., 2025; Razin et al., 2024; Ren & Sutherland, 2024), the stagnation or reduction of likelihood for both correct and incorrect responses during GRPO optimization (Deng et al., 2025), as the fundamental but

*previously overlooked* source of collapse in TIRL. Using search-integrated QA as a case study (see Figure 2), we show that LLD emerges early and persistently: even as rewards increase, the likelihood of correct responses enters a monotonic decline. This behavior appears across model scales, indicating that LLD is a structural failure mode of GRPO-based TIRL rather than a configuration-specific artifact. We further demonstrate that LLD drives a self-reinforcing LLD death spiral, where the resulting low-confidence regime amplifies negative-gradient influence from incorrect trajectories, accelerating likelihood decay, triggering entropy spikes, inflating likelihood ratios, and ultimately causing the large-gradient instability that leads to collapse.

To counteract this failure mode, we propose a simple yet effective LLD suppression (LLDS) regularization that prevents harmful likelihood reductions. Our method integrates seamlessly with GRPO and introduces two layers of selectivity: (i) *action-level gating*, which activates the regularization only when a response action's overall likelihood decreases, and (ii) *token-level selectivity*, which penalizes only the tokens responsible for the decrease. This fine-grained design directly mitigates LLD while minimally interfering with GRPO's optimization behavior. By preventing unintentional downward likelihood drift, LLDS maintains stable training, suppressed gradient explosion, and consistent gains across seven open-domain and multi-hop QA benchmarks. Our main contributions are threefold:

• We identify LLD as a previously unrecognized failure mode in GRPO-based tool-integrated reinforcement learning, and show that it arises pervasively and follows a consistent, self-reinforcing trajectory of sustained likelihood decay, gradient amplification, entropy explosion, and eventual training collapse.

• We propose a lightweight regularization LLDS that selectively regularize likelihood reductions and resolves the collapse issue in GRPO training.

• By stabilizing training, we realize significant performance gains across QA benchmarks (see Figure 1), demonstrating a more robust and reliable approach to TIRL.

## 2. Related Work

**Tool-Integrated Reasoning and Agentic LLMs.** Tool use enables LLMs to perform adaptive, multi-step reasoning. Early methods relied on prompt-based (Lu et al., 2023; Shen et al., 2023) or supervised tool calling (Gou et al., 2023; Qin et al., 2023), while recent RL-based systems (e.g., RETool (Feng et al., 2025), VERL-Tool (Jiang et al., 2025)) learn tool-usage policies from environmental feedback, achieving strong performance across QA (Jin et al., 2025), code-based math reasoning (Xue et al., 2025), SQL generation (Jiang et al., 2025), and multimodal tasks (Gao et al., 2024).

**Training Collapse in Tool-Integrated GRPO.** GRPO (Guo et al., 2025) is a value-free, outcome-driven RL method but exhibits severe instability in multi-turn tool-integrated settings, often collapsing when trained from base models with only verifiable rewards (Jin et al., 2025). This behavior is consistently observed in systems such as Search-R1 (Jin et al., 2025), SimpleTIR (Xue et al., 2025), and ZeroSearch (Sun et al., 2025), whereas PPO remains comparatively stable. Prior explanations attribute the collapse to training–inference mismatch (Liu et al., 2025) or low-likelihood incorrect responses (Xue et al., 2025), but do not explain PPO's robustness or the structural origin of the instability. We identify Lazy Likelihood Displacement (LLD) as the underlying mechanism driving collapse in multi-turn TIRL (Section B.1). Recent works mitigate this via turn-level reward shaping, either relying on ground-truth documents (Zheng et al., 2025b) or an external LLM judge (Zhang et al., 2025), both introducing additional supervision or system cost and increased vulnerability to reward hacking (Guo et al., 2025). Recently, TreeGRPO (Ji et al., 2025) provides dense reward using tree-based rollout, but increases rollout and computational overhead. In contrast, our approach shows that outcome rewards alone suffice to achieve superior performance once GRPO is stabilized. Detailed related work see Section B.1.

## 3. Preliminary

We denote the query by $\boldsymbol{x}$, the tool feedback by $\boldsymbol{o}$, and the model's action by $\boldsymbol{y}$. For a given query $\boldsymbol{x}$, the model autoregressively generates an action at turn $t$ according to

$$\pi_\theta(\boldsymbol{y}_t \mid \boldsymbol{x}, \boldsymbol{y}_0, \boldsymbol{o}_0, \boldsymbol{y}_1, \ldots, \boldsymbol{o}_{t-1}),$$

where $\boldsymbol{y}_t$ is produced based on the accumulated context, and $\boldsymbol{o}_{t-1} \sim \mathcal{T}$ denotes the tool feedback returned at turn $t-1$ if the previous action $\boldsymbol{y}_{t-1}$ invoked a tool $\mathcal{T}$ (e.g., issuing a search query). If no tool is invoked, $\boldsymbol{o}_{t-1}$ is an empty string. The *multi-turn* setting thus corresponds to trajectories involving multiple tool calls. However, tool feedback $\boldsymbol{o}_t$ is inherently out-of-distribution (OOD) for the LLM, as it originates from external environments rather than the model's generative distribution. Thus, prior work (Jin et al., 2025) masks feedback tokens during training to stabilize optimization.

**Tool-Integrated GRPO with Feedback Mask.** GRPO, introduced in DeepSeek-Math (Shao et al., 2024) and DeepSeek-R1 (Guo et al., 2025) and later applied to tool use (Jin et al., 2025), is a value-free policy optimization method with group-relative reward normalization. Specifically, for a query–answer pair $(\boldsymbol{x}, \boldsymbol{a})$, the policy $\pi_\theta$ samples $G$ responses:

$$\{(\boldsymbol{y}_{i,0}, \boldsymbol{o}_{i,0}, \ldots, \boldsymbol{y}_{i,t}, \boldsymbol{o}_{i,t}, \ldots, \boldsymbol{o}_{i,T_i-1}, \boldsymbol{y}_{i,T_i})\}_{i=1}^G,$$

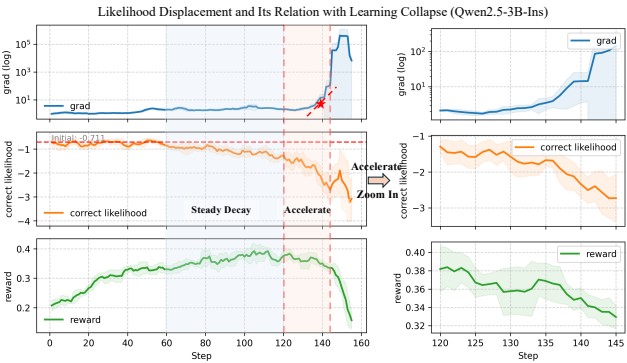

*Figure 2.* We illustrate the likelihood displacement in tool-integrated RL training. The steady-decay phase (60-120) emerges even when the reward increases. In the subsequent acceleration phase (after step 120), the likelihood of correct responses drops sharply, accompanied by a sudden surge in gradient magnitude (red star), leading to gradient explosion. A zoomed-in view of the acceleration region further highlights this effect, showing a clearer likelihood displacement, where the gradient accelerates rapidly while the reward starts to decline.

where $T_i$ denotes the number of tool calls, and thus determines the number of multi-turn interactions in the $i$-th rollout. Each action $\boldsymbol{y}_{i,t}$ consists of $|\boldsymbol{y}_{i,t}|$ tokens, and we denote by $\boldsymbol{y}_{i,t,<k}$ the prefix consisting of its first $k-1$ tokens. Let $r_i$ denote the reward assigned to the $i$-th response. The advantage for the $t$-th action of $i$-th response is defined via group-level normalization:

$$\hat{A}_{i,t,k} := \frac{r_i - \mu}{\sigma}, \qquad k = 1, \ldots, |\hat{\boldsymbol{y}}_{i,t}|,$$

where $\mu = \widehat{\mathbb{E}}[\{r_i\}_{i=1}^G]$ and $\sigma = \sqrt{\widehat{\text{Var}}[\{r_i\}_{i=1}^G]}$ are the empirical mean and standard deviation of rewards within the group. Each token of the same trajectory shares the same normalized advantage. The tool-integrated GRPO objective **with feedback mask** is given by:

$$\mathcal{J}_{\text{GRPO}}(\theta) = \mathbb{E}_{\substack{(\boldsymbol{x},\boldsymbol{a})\sim\mathcal{D} \\ \{(\boldsymbol{y}_{i,t},\boldsymbol{o}_{i,t})_{t=0}^{T_i-1},\boldsymbol{y}_{i,T_i}\}\sim(\pi_{\theta_{\text{old}}},\mathcal{T})}}$$

$$\left[ \frac{1}{\sum_{i=1}^G \sum_{t=1}^{T_i} |\hat{\boldsymbol{y}}_{i,t}|} \sum_{i=1}^G \sum_{t=1}^{T_i} \sum_{k=1}^{|\hat{\boldsymbol{y}}_{i,t}|} \min\left(\gamma_{i,t,k}(\theta)\,\hat{A}_{i,t,k}, \right.\right.$$

$$\left.\left. \hat{A}_{i,t,k} \, \text{clip}\left(\gamma_{i,t,k}(\theta),\, 1-\varepsilon,\, 1+\varepsilon\right)\right)\right], \tag{1}$$

where $\varepsilon$ is the clipping parameter, the likelihood ratio is defined as $\gamma_{i,t,k}(\theta) = \frac{\pi_\theta(\boldsymbol{y}_{i,t,k}|\boldsymbol{x},\ldots,\boldsymbol{y}_{i,t-1},\boldsymbol{o}_{i,t-1},\boldsymbol{y}_{i,t,<k})}{\pi_{\theta_{\text{old}}}(\boldsymbol{y}_{i,t,k}|\boldsymbol{x},\ldots,\boldsymbol{y}_{i,t-1},\boldsymbol{o}_{i,t-1},\boldsymbol{y}_{i,t,<k})}$. Although feedback tokens $\boldsymbol{o}$ are masked out, they still influence the context of subsequent token predictions.

## 4. Lazy Likelihood Displacement in Tool-integrated GRPO

We identify a critical but previously unrecognized failure mode in tool-integrated GRPO, in which likelihood dis-

placement progressively accelerates through compounding low-confidence responses and ultimately destabilizes training. While (Deng et al., 2025) identifies LLD as a training pathology in single-turn GRPO, their analysis is restricted to closed-loop text generation without external observations. In contrast, tool-integrated reinforcement learning fundamentally changes the learning dynamics: LLD becomes a trajectory-level instability that interacts with out-of-distribution tool feedback and prefix-shared decision branches, leading to systematic likelihood decay, gradient amplification, and eventual training collapse.

We further show that this phenomenon arises from two structural factors unique to the tool-integrated GRPO setting (analyzed in Section 5.4) and consistently leads to catastrophic collapse behavior. We formalize and define LLD in the tool-integrated RL regime in this section.

**Definition 4.1** (Tool-LLD). Let $\pi_{\theta_{\text{old}}}$ and $\pi_{\theta_{\text{fin}}}$ denote the initial and finetuned policies obtained before and after optimizing the objective $\mathcal{J}$ (e.g., Equation (1)) over a dataset $\mathcal{D}$, with $\mathcal{J}(\theta_{\text{fin}}) < \mathcal{J}(\theta_{\text{old}})$. Consider a tool-integrated trajectory consisting of alternating actions and feedback, $(\boldsymbol{y}_0, \boldsymbol{o}_0, \boldsymbol{y}_1, \boldsymbol{o}_1, \ldots, \boldsymbol{y}_T)$, where only the actions $\{\boldsymbol{y}_t\}_{t=0}^T$ are used in likelihood computation (feedback is masked). For each response action $\boldsymbol{y}_t$, define its log-likelihood change as

$$\Delta_t(\mathbf{x}, \boldsymbol{y}_t) := \ln \pi_{\theta_{\text{fin}}}(\boldsymbol{y}_t \mid \mathbf{x}, \boldsymbol{y}_{<t}, \boldsymbol{o}_{<t}) - \ln \pi_{\theta_{\text{old}}}(\boldsymbol{y}_t \mid \mathbf{x}, \boldsymbol{y}_{<t}, \boldsymbol{o}_{<t}).$$

We say that LLD occurs for the action $t$ if $\Delta_t(\mathbf{x}, \boldsymbol{y}_t) \leq \epsilon$, where $\epsilon$ is a small or non-positive constant. And LLD occurs for the whole response if

$$\sum_{t=0}^T \Delta_t(\mathbf{x}, \boldsymbol{y}_t) \leq \epsilon, \tag{2}$$

Therefore, LLD accounts for the failure scenario where **even correct response actions fail to boost confidence (or even reduce it)** when such an enhancement is explicitly required by the loss function.

We generalize the negative-gradient characterization of group-relative updates in (Deng et al., 2025) to a multi-turn setting with external observations, where the learning dynamics differ qualitatively. We present the following informal theorem (Formal statement and proof in Section A).

**Theorem 4.2** (Informal: Trajectory-Level LLD in Tool-Integrated GRPO). *In tool-integrated GRPO, the likelihood of a correct response action can decrease when incorrect responses satisfy two conditions:* (i) Low likelihood: *The model assigns low probability to incorrect responses, causing large prediction-error weights that magnify their influence, and* (ii) High embedding similarity: *Incorrect responses share similar representations with correct ones,*

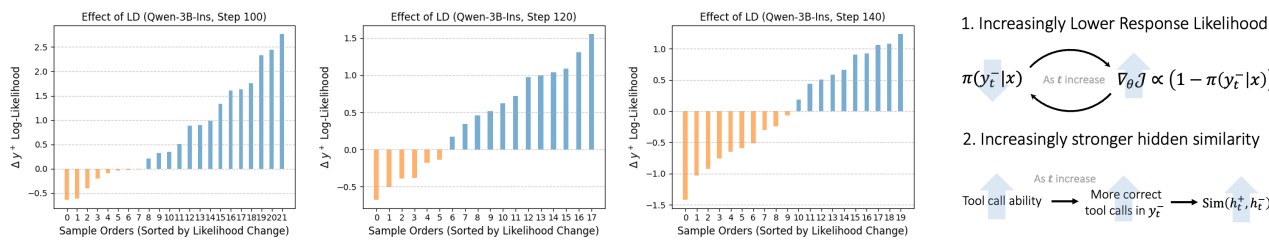

*(a)* Likelihood displacement across training steps.          *(b)* Mechanism

*Figure 3.* (a) Effect of likelihood displacement across different training iterations for the Qwen2.5-3B-Instruct model. Results are computed on the first 50 samples of the training set, discarding cases where all responses are uniformly correct or uniformly incorrect. Bars below zero (orange) indicate samples whose correct responses' likelihood decreases after training. (b) Illustration of the mechanism: as training progresses, the likelihood of correct responses decreases while hidden-state similarity increases, leading to stronger LLDS.

*causing negative gradients to interfere with positive updates. As they frequently do in tool-integrated settings (see Section 5.4), negative gradients dominate, causing LLD.*

In the setting of tool-integrated GRPO, we observed **with striking frequency** that the likelihood of correct responses reduces (Figure 2). This indicates an especially acute form of LLD ($\epsilon \leq 0$), causing a **progressive decay in the model's output likelihood**. This compounding decay is a distinctive failure mode of Search-R1 style TIRL.

## 5. LLD in Tool-Integrated GRPO Collapse

This section examines the prevalence of LLD in tool-integrated GRPO and illustrates how sustained likelihood decay ($\epsilon \leq 0$) leads to catastrophic training collapse. All analyses in this section use the NQ dataset (Kwiatkowski et al., 2019) unless otherwise stated.

### 5.1. Likelihood Dynamic

To characterize the prevalence and evolution of LLD during training, we visualize the training trajectory in Figure 2. The dynamics exhibit three distinct phases. **Phase I (early stagnation).** The likelihood of correct responses remains nearly constant despite increasing rewards, indicating the onset of LLD. The subsequent phases correspond to the $\epsilon \leq 0$ regime, where likelihood strictly decreases. **Phase II (steady decay).** Likelihood declines slowly but consistently, while rewards continue to rise and gradient norms remain stable (e.g., steps 60–120 in Figure 2). **Phase III (acceleration).** After a turning point (around step 120 in Figure 2), likelihood drops sharply and coincides with a rapid increase in gradient magnitude (marked by the red star), leading to gradient explosion and eventual training collapse. We further provide a zoomed-in view showing that, although collapse is ultimately triggered by gradient explosion, instability and reward degradation emerge earlier.

### 5.2. LLD Death Spiral

In Figure 2, we observe an accelerated decline in response likelihood as training progresses. We formalize the resulting accelerated likelihood decay as an LLD death spiral.

**Definition 5.1** (Informal LLD Death Spiral). Consider a policy update from $\pi_{\theta_s}$ to $\pi_{\theta_{s+1}}$. We say the system enters an *LLD Death Spiral* if the policy evolution follows the self-reinforcing pattern

$$\text{LLD}_s \implies C_s^{\text{low}} \implies \text{LLD}_{s+1}, \qquad \varepsilon_{s+1} < \varepsilon_s \leq 0,$$

where $\text{LLD}_s$ denotes lazy likelihood displacement at iteration $s$, and $C_s^{\text{low}}$ denotes low-confidence trajectories whose likelihood decreases under $\pi_{\theta_s}$.

A formal definition is provided in Section A.2, and the mechanism is visualized in Figure 3. Beyond the acceleration phase shown in Figure 2, we further present additional empirical evidence of the LLD death spiral in the following section.

### 5.3. Empirical Evidence for LLD Death Spiral

We demonstrate the LLD Death Spiral through *accelerated entropy explosion* and *accelerated per-sample LLD*.

**Accelerated Per-Sample LLD.** We conduct a controlled study on the NQ dataset (Kwiatkowski et al., 2019) to examine how GRPO training affect the likelihood of correct responses. Using Qwen2.5-3B-Ins (Yang et al., 2024), we generate eight rollouts per question and retain only samples containing both correct and incorrect responses. For each question, we reinitialize model parameters, apply a single GRPO update, and measure the average log-likelihood change of correct responses:

$$\Delta(\boldsymbol{x}) := \frac{1}{N^+} \sum_{i=1}^{N^+} \sum_{t=0}^{T_i} \left[ \ln \pi_{\theta'}(\boldsymbol{y}_{i,t}^+ \mid \boldsymbol{x}) - \ln \pi_{\theta}(\boldsymbol{y}_{i,t}^+ \mid \boldsymbol{x}) \right], \tag{3}$$

where $N^+$ denotes the number of correct responses. As shown in Figure 3a, likelihood degradation emerges as training progresses (orange curve). During the acceleration phase (iterations 120–140), this effect intensifies into an *LLD death spiral*, with $> 50\%$ samples exhibiting likelihood drops.

**Accelerated Entropy Explosion.** Figure 4 shows the evolution of entropy, response length, and valid-search ratio for

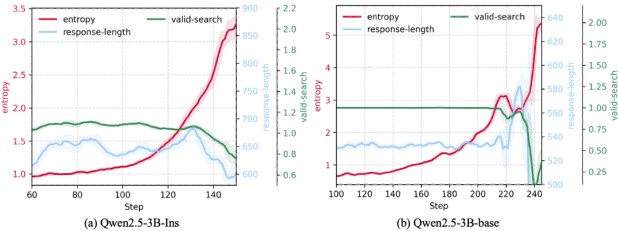

*Figure 4.* We visualize the evolution of entropy, response length, and valid-search ratio during training. For both Qwen2.5-3B-Instruct and Qwen2.5-3B-Base, entropy rises sharply prior to collapse, indicating severe likelihood displacement.

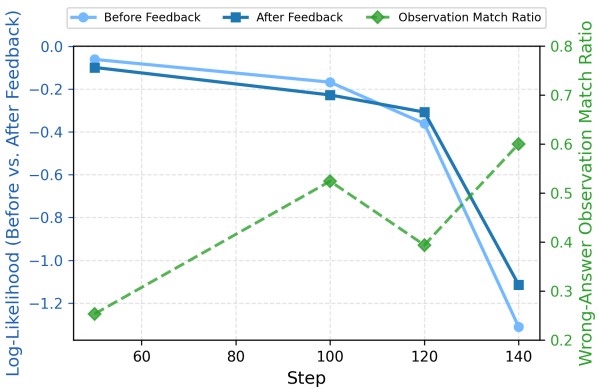

*Figure 5.* Evolution of token log-likelihood (measured before vs. after feedback; left axis) and the observation-match ratio for wrong answers (right axis). With training, both likelihoods drop while the overlap between tool observations in incorrect and correct trajectories increases, suggesting many incorrect responses begin with a correct search, which skews likelihood estimates and contributes to LLD.

Qwen2.5-3B-Ins and Qwen2.5-3B-Base. For both models, token entropy increases slowly in early training, consistent with the slow-LLD regime in Figure 2, and then accelerates sharply during the LLD acceleration phase. This entropy surge reflects the LLD death spiral: accumulating low-confidence responses induce increasingly diffuse token distributions, further amplifying likelihood decay and instability. Notably, response length and valid-search frequency remain nearly constant, indicating that the entropy growth and likelihood degradation are driven by LLD rather than longer trajectories or increased tool usage.

### 5.4. Why Tool-Integrated (TI) GRPO Amplifies LLD

We then show that the severity of LLD in TI-GRPO stems from two interacting factors predicted by Theorem 4.2 and demonstrated in Figure 3b: high response similarity between correct and incorrect trajectories, and low likelihood responses that induce large prediction-error weights.

**Frequent correct actions in incorrect responses.** Our analysis on Qwen2.5-3B-Ins that trained on the NQ dataset reveals a striking pattern: correct actions frequently appear

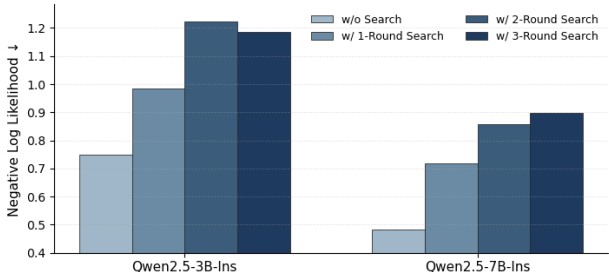

*Figure 6.* Effect of off-policy tool context on negative log-likelihood. Increasing the number of tool-interaction rounds consistently reduces response likelihood.

within otherwise incorrect responses. Specifically, in TI GRPO, the first response action typically formulates a search query, whose correctness increases steadily during training regardless of final answer. We label the first action of incorrect responses as correct if its retrieved documents match those of a correct response. As shown in Fig. 5, this accuracy (green line) is initially low but rises sharply, reaching ~60% by step 140, indicating high similarity on the first action between correct and incorrect trajectories. At the same time, the likelihood of the first action (light blue) decays faster than that of the second action (blue): although initially higher due to the second action conditioning on OOD feedback, it drops below the second action around step 110, where first-action correctness reaches ~50%. After step 120, this decay accelerates sharply, where low-likelihood incorrect prefixes further amplify degradation. As LLD intensifies, the likelihood of the first action collapses, and the model begins producing random, nonsensical tokens (Appendix, Fig. 15), rendering responses effectively unusable even before full training collapse. These results motivate mitigating unintended penalization of correct actions within incorrect trajectories (see Section D).

**Low Likelihood reasoning after tool feedback.** A second factor that amplifies LLD is the use of OOD tool feedback as a conditioning context. They introduce a persistent distribution mismatch between the policy and its input context, systematically reducing the likelihood assigned to subsequent actions and increasing negative log-likelihood. As shown in Figure 6, computed on the first 50 NQ training samples, this effect strengthens with the number of tool-interaction rounds, producing a monotonic increase in NLL for both Qwen2.5-3B-Instruct and Qwen2.5-7B-Instruct models.

### 5.5. LLD Suppression regularization

To mitigate LLD, we propose a likelihood-preserving regularizer, LLDS, which prevents unintended reductions in response likelihood during GRPO training. Specifically, for each given preserving response $y_i \in \mathcal{Y}_{\text{pre}}$, we compare token-level likelihoods under the previous and current

policies, and quantify their change by

$$\Delta_{i,t,k} = \ln \pi_{\theta_{\mathrm{old}}}(y_{i,t,k} \mid \mathbf{x}, \boldsymbol{y}_{i,<t}, \boldsymbol{o}_{i,<t}, \boldsymbol{y}_{i,t,<k})$$
$$- \ln \pi_\theta(y_{i,t,k} \mid \mathbf{x}, \boldsymbol{y}_{i,<t}, \boldsymbol{o}_{i,<t}, \boldsymbol{y}_{i,t,<k}).$$

where $i$ indexes samples, $t$ actions (turns), and $k$ tokens. Positive $\Delta_{i,t,k}$ indicates a likelihood decrease after the update. These tokens are candidates for regularization.

**LLDS: Action-Level Gating.** To avoid unnecessary penalties on globally improving responses, we introduce an *action-level gating* mechanism: the penalty activates only when the *total* likelihood of an action decreases. The resulting LLDS loss is defined as:

$$L_{\mathrm{LLDS}} = \frac{1}{\sum_{i=1}^{|\mathcal{Y}_{\mathrm{pre}}|} \sum_{t=0}^{T_i} |\boldsymbol{y}_{i,t}|} \sum_{i=1}^{|\mathcal{Y}_{\mathrm{pre}}|} \sum_{t=0}^{T_i}$$
$$\underbrace{\mathbb{1}\left[\sum_{k=1}^{|\boldsymbol{y}_{i,t}|} \Delta_{i,t,k} > 0\right]}_{\text{Activated only when sum} > 0} \sum_{k=1}^{|\boldsymbol{y}_{i,t}|} \underbrace{\max(0, \Delta_{i,t,k})}_{\text{Likelihood-reducing tokens}}, \quad (4)$$

This structure directly suppresses actions that suffer from LLD. We also introduce token-level and response-level variants in Section B.3, unless stated otherwise, LLDS refers to the action-level gating.

**LLDS-MA: Masking Answer Tokens.** To further promote multi-turn reasoning and tool usage, we assign smaller regularization weights to final answer tokens, discouraging premature answer generation. Consequently, LLDS-MA penalizes likelihood reductions on reasoning and tool-interaction tokens more strongly, while only softly regularizing the answer span $\boldsymbol{y}_{i,\mathrm{Ans}}$:

$$\sum_{k=1}^{|\boldsymbol{y}_{i,t}|} w_{i,t,k} \max(0, \Delta_{i,t,k}), w_{i,t,k} = \begin{cases} \beta, & k \in \boldsymbol{y}_{i,\mathrm{Ans}} \\ 1, & \text{otherwise} \end{cases} \quad (5)$$

Finally, we integrate the regularization into the GRPO objective as

$$L_{\mathrm{total}} = L_{\mathrm{GRPO}} + \lambda L_{\mathrm{LLDS(-MA)}} \quad (6)$$

where $\lambda$ is the regularization weight. We use LLDS as the default variant, and adopt LLDS-MA when the model collapses to single-turn behavior. The preserving set $\mathcal{Y}_{\mathrm{pre}}$ includes all responses with non-negative advantages $\hat{A} \geq 0$, ensuring that correct responses ($\hat{A} > 0$) and untrained responses ($\hat{A} = 0$) do not suffer LLD. The effect of $\lambda$ is examined empirically in Figure 8.

# 6. Experiments and Analysis

**Experimental settings.** For training, we follow the setup in (Jin et al., 2025) and conduct experiments using two

model families: Qwen-2.5-3B and Qwen-2.5-7B, each in both Base and Instruct variants (Yang et al., 2024). We follow the training configuration in Search-R1 (Jin et al., 2025) and use **NQ+Hotpot (single-hop+multi-hop)**: the model is trained on a merged corpus combining NQ with HotpotQA (Yang et al., 2018), providing broader coverage of both open-domain and multi-hop reasoning. For the retrieval dataset, we use the 2018 Wikipedia dump (Karpukhin et al., 2020) as the knowledge corpus and employ E5 (Wang et al., 2022) as the dense retriever and fix the number of retrieved passages to three. Unless otherwise specified, we use the same optimization hyperparameters as Search-R1 (Jin et al., 2025), with the only modification being a reduced maximum turn from 4 to 3 for the NQ+Hotpot, improving training efficiency. Unless stated otherwise, the regularization weight is fixed at $\lambda = 0.2$. More details see Sec. B.2.1.

**Evaluation settings.** The Evaluation is performed on seven datasets, categorized as follows: (1) *General QA:* NQ (Kwiatkowski et al., 2019), TriviaQA (Joshi et al., 2017), and PopQA (Mallen et al., 2022). (2) *Multi-Hop QA:* HotpotQA (Yang et al., 2018), 2WikiMultiHopQA (Ho et al., 2020), Musique (Trivedi et al., 2022), and Bamboogle (Press et al., 2022). We follow (Jin et al., 2025) and adopt exact match (EM) as the evaluation metric.

**Baseline Methods.** To evaluate the effectiveness of LLDS, we use baselines in Search-R1 (Jin et al., 2025) and additionally include methods that improve training stability via dense rewards, namely StepSearch (Zheng et al., 2025b), CriticSearch (Zhang et al., 2025), and TreeGRPO (Ji et al., 2025). See Section B.2.2 for detailed descriptions.

## 6.1. Experimental Results

We evaluate our proposed LLDS on seven open-domain and multi-hop QA benchmarks using two model families: Qwen2.5-3B (Base/Instruct) and Qwen2.5-7B (Base/Instruct). Tables 1 and 2 summarize the EM performance across all settings. Since vanilla GRPO training often collapses, we use the results from Search-R1 (Jin et al., 2025), corresponding to the best checkpoint before collapse, as the GRPO baseline. This choice favors GRPO and avoids underestimating its performance. More results see Section B.4.

**Results on Qwen2.5-3B.** As shown in Table 1, LLDS improves the vanilla GRPO score from 0.312 to 0.360, corresponding to a 15.4% relative gain. Notably, because the 3B base model tends to invoke search only once, we apply LLDS-MA, which achieves the best performance with an average score of 0.440, representing a substantial +45.2% improvement over GRPO. A similar trend is observed for Qwen2.5-3B-Instruct, where LLDS attains 0.441 under the NQ+Hotpot setting (+31.3%).

**Results on Qwen2.5-7B.** As shown in Table 2, LLDS delivers substantial gains over vanilla GRPO for both base

| Methods | General QA | | | Gen-Avg | Multi-Hop QA | | | | MH-Avg | Avg |
|---|---|---|---|---|---|---|---|---|---|---|
| | NQ[†] | TriviaQA[*] | PopQA[*] | | HotpotQA[†] | 2Wiki[*] | Musique[*] | Bamboogle[*] | | |
| **Qwen2.5-3b-Base/Instruct** | | | | | | | | | | |
| Direct Inference[◇] | 0.106 | 0.288 | 0.108 | 0.167 | 0.149 | 0.244 | 0.020 | 0.024 | 0.109 | 0.134 |
| Search-o1[◇] | 0.238 | 0.472 | 0.262 | 0.324 | 0.221 | 0.218 | 0.054 | 0.320 | 0.203 | 0.255 |
| RAG[◇] | 0.348 | 0.544 | 0.387 | 0.426 | 0.255 | 0.226 | 0.047 | 0.080 | 0.152 | 0.270 |
| R1-base[◇] | 0.226 | 0.455 | 0.173 | 0.285 | 0.201 | 0.268 | 0.055 | 0.224 | 0.187 | 0.229 |
| R1-instruct[◇] | 0.210 | 0.449 | 0.171 | 0.277 | 0.208 | 0.275 | 0.060 | 0.192 | 0.184 | 0.224 |
| Rejection Sampling[◇] | 0.294 | 0.488 | 0.332 | 0.371 | 0.240 | 0.233 | 0.059 | 0.210 | 0.186 | 0.265 |
| Search-R1-PPO-Base[◇] | 0.406 | 0.587 | 0.435 | 0.476 | 0.284 | 0.273 | 0.049 | 0.088 | 0.174 | 0.303 |
| Search-R1-PPO-Ins[◇] | 0.341 | 0.545 | 0.378 | 0.421 | 0.324 | 0.319 | 0.103 | 0.264 | 0.253 | 0.325 |
| ZeroSearch-Base | 0.430 | 0.616 | 0.414 | 0.487 | 0.338 | 0.346 | 0.130 | 0.139 | 0.238 | 0.345 |
| ZeroSearch-Ins | 0.414 | 0.574 | 0.448 | 0.479 | 0.274 | 0.300 | 0.098 | 0.111 | 0.196 | 0.317 |
| StepSearch | 0.296 | 0.490 | 0.341 | 0.375 | 0.294 | 0.360 | 0.140 | 0.258 | 0.263 | 0.311 |
| CriticSearch[△] | - | - | - | - | 0.414 | 0.409 | 0.180 | 0.368 | 0.343 | - |
| Tree-GRPO | 0.468 | 0.597 | 0.436 | 0.500 | 0.424 | 0.437 | 0.178 | 0.432 | 0.368 | 0.424 |
| *Qwen2.5-3b-Base* | | | | | | | | | | |
| Search-R1-GRPO[◇] | 0.421 | 0.583 | 0.413 | 0.472 | 0.297 | 0.274 | 0.066 | 0.128 | 0.191 | 0.312 |
| +LLDS | 0.479 | 0.627 | 0.453 | 0.520 | 0.345 | 0.323 | 0.082 | 0.210 | 0.240 | 0.360 |
| +LLDS-MA | **0.483** | **0.633** | **0.474** | **0.530** | **0.452** | **0.443** | **0.197** | **0.395** | **0.372** | **0.440** (+45.2%) |
| *Qwen2.5-3b-Ins* | | | | | | | | | | |
| Search-R1-GRPO[◇] | 0.397 | 0.565 | 0.391 | 0.451 | 0.331 | 0.310 | 0.124 | 0.232 | 0.249 | 0.336 |
| +LLDS | **0.462** | **0.627** | **0.468** | **0.519** | **0.443** | **0.447** | **0.197** | **0.444** | **0.383** | **0.441** (+31.3%) |
| Search-R1-GSPO | 0.376 | 0.561 | 0.374 | 0.437 | 0.332 | 0.329 | 0.098 | 0.250 | 0.252 | 0.331 |
| +LLDS | **0.489** | **0.640** | **0.483** | **0.537** | **0.466** | **0.456** | **0.215** | **0.403** | **0.385** | **0.451** (+36.3%) |

*Table 1.* Results on General QA and Multi-Hop QA datasets for **Qwen2.5-3b-Base/Instruct**. All results are reported using Exact Match (EM). [◇] denotes results from (Jin et al., 2025). [△] denotes results from (Zhang et al., 2025) that only have multi-hop QA results. [†] denotes in-distribution datasets, while [*] denotes out-of-distribution datasets.

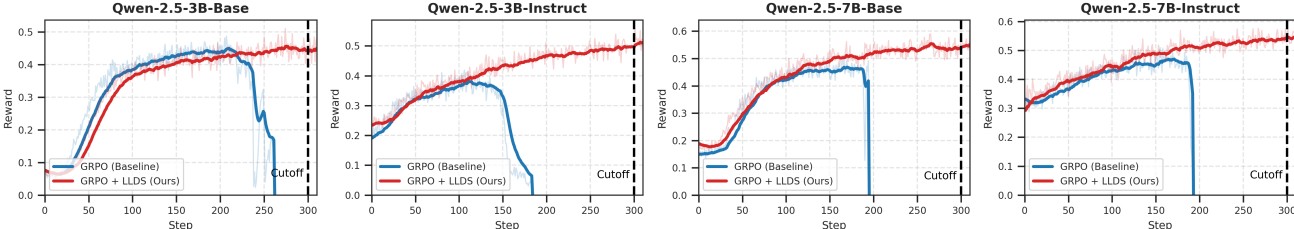

*Figure 7.* Comparisons of training reward curves between baseline GRPO (blue) and GRPO + LLDS (red). From left to right: Qwen-2.5-3B-Base, Qwen-2.5-3B-Instruct, Qwen-2.5-7B-Base, and Qwen-2.5-7B-Instruct. The baseline consistently collapses (reward drops to zero), while LLDS stabilizes training and sustains high rewards.

and instruct variants. For Qwen2.5-7B-Base, it improves the average score from 0.350 to 0.480, a 37.1% relative increase, while for Qwen2.5-7B-Instruct, performance rises from 0.396 to 0.483, corresponding to a 22.0% gain. Notably, LLDS attains the best results on all Multi-Hop QA benchmarks with MH-Avg accuracy being 0.426.

**Comparison with Dense Rewards.** As shown in Tables 1 and 2, stabilizing outcome-reward GRPO training with LLDS consistently outperforms methods that rely on dense reward supervision. These dense-reward approaches typically require turn-level ground-truth signals, external LLM judges, or more expensive tree-based rollouts. In contrast, LLDS achieves superior performance using only outcome rewards. Notably, LLDS outperforms the strongest dense-reward baseline, Tree-GRPO, by 4% and 10% on the 3B and 7B models, respectively, and exceeds other dense-reward

methods by over 30%, demonstrating both improved effectiveness and substantially lower supervision.

**Results on GSPO.** We further show that LLDS integrates seamlessly with other group-relative reinforcement learning objectives. In particular, we apply LLDS to Group Sequence Policy Optimization (GSPO) (Zheng et al., 2025a), which optimizes sequence-level importance ratios. As shown in Table 1, vanilla GSPO also suffers from limited performance due to training collapse, whereas incorporating LLDS consistently improves results across both General QA and Multi-Hop QA benchmarks. Notably, GSPO+LLDS achieves the highest overall average score (0.451) on Qwen2.5-3B-Instruct, outperforming GSPO by 36.3%. These results confirm the generality of LLDS across group-relative optimization formulations.

| Methods | General QA | | | Gen-Avg | Multi-Hop QA | | | | MH-Avg | Avg |
|---|---|---|---|---|---|---|---|---|---|---|
| | NQ[†] | TriviaQA* | PopQA* | | HotpotQA[†] | 2Wiki* | Musique* | Bamboogle* | | |
| **Qwen2.5-7b-Base/Instruct** | | | | | | | | | | |
| Direct Inference$^\diamond$ | 0.134 | 0.408 | 0.140 | 0.227 | 0.183 | 0.250 | 0.031 | 0.120 | 0.146 | 0.181 |
| Search-o1$^\diamond$ | 0.151 | 0.443 | 0.131 | 0.242 | 0.187 | 0.176 | 0.062 | 0.296 | 0.180 | 0.206 |
| RAG$^\diamond$ | 0.349 | 0.585 | 0.392 | 0.442 | 0.299 | 0.235 | 0.058 | 0.208 | 0.200 | 0.304 |
| R1-base$^\diamond$ | 0.297 | 0.539 | 0.199 | 0.345 | 0.242 | 0.273 | 0.083 | 0.203 | 0.200 | 0.262 |
| R1-instruct$^\diamond$ | 0.270 | 0.537 | 0.199 | 0.335 | 0.237 | 0.292 | 0.072 | 0.293 | 0.224 | 0.271 |
| Rejection Sampling$^\diamond$ | 0.360 | 0.592 | 0.380 | 0.444 | 0.331 | 0.296 | 0.123 | 0.355 | 0.276 | 0.348 |
| Search-R1-PPO-Base$^\diamond$ | 0.480 | 0.638 | 0.457 | 0.525 | 0.433 | 0.382 | 0.196 | 0.432 | 0.361 | 0.431 |
| Search-R1-PPO-Ins$^\diamond$ | 0.393 | 0.610 | 0.397 | 0.467 | 0.370 | 0.414 | 0.146 | 0.368 | 0.325 | 0.385 |
| ZeroSearch-Base | 0.424 | 0.664 | 0.604 | 0.564 | 0.320 | 0.340 | 0.180 | 0.333 | 0.293 | 0.409 |
| ZeroSearch-Ins | 0.436 | 0.652 | 0.488 | 0.525 | 0.346 | 0.352 | 0.184 | 0.278 | 0.290 | 0.391 |
| StepSearch | 0.364 | 0.565 | 0.390 | 0.440 | 0.346 | 0.394 | 0.183 | 0.444 | 0.342 | 0.382 |
| CriticSearch$^\triangle$ | - | - | - | - | 0.442 | 0.428 | 0.194 | 0.472 | 0.384 | - |
| Tree-GRPO | 0.481 | 0.633 | 0.452 | 0.522 | 0.446 | 0.423 | 0.202 | 0.440 | 0.378 | 0.439 |
| *Qwen2.5-7b-Base* | | | | | | | | | | |
| Search-R1-GRPO$^\diamond$ | 0.395 | 0.560 | 0.388 | 0.448 | 0.326 | 0.297 | 0.125 | 0.360 | 0.277 | 0.350 |
| +LLDS | **0.512** | **0.672** | **0.480** | **0.555** | **0.486** | **0.469** | **0.232** | **0.508** | **0.424** | **0.480** (+37.1%) |
| *Qwen2.5-7b-Ins* | | | | | | | | | | |
| Search-R1-GRPO$^\diamond$ | 0.429 | 0.623 | 0.427 | 0.493 | 0.386 | 0.346 | 0.162 | 0.400 | 0.324 | 0.396 |
| +LLDS | **0.518** | **0.676** | **0.486** | **0.560** | **0.492** | **0.490** | **0.228** | **0.492** | **0.426** | **0.483** (+22.0%) |

*Table 2.* Performance comparison on General QA and Multi-Hop QA datasets for **Qwen2.5-7b-Base/Instruct**.

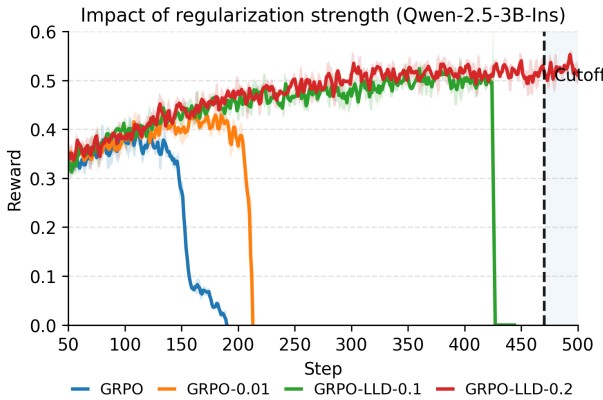

*Figure 8.* Impact of regularization strength in training Qwen2.5-3B-Instruct on NQ and HotpotQA.

## 6.2. Ablation Study and Analysis

**Impact of Regularization Strength.** We ablate $\lambda \in \{0, 0.01, 0.1, 0.2\}$ and plot the corresponding training reward dynamics in Figure 8. As shown, Vanilla GRPO ($\lambda = 0$) collapses around step 200. A small regularization value ($\lambda = 0.01$, orange curve) delays but does not prevent collapse, which occurs around step 220. In contrast, a stronger regularization ($\lambda = 0.2$) stabilizes training entirely, allowing the model to continue smoothly without collapse. We truncate the plot at step 500 for clarity, although training can proceed well beyond this point.

**Impact of masking answer (MA).** We apply MA only to model variants where vanilla GRPO degenerate into issuing only a single search call. MA reduces LLDS regularization on the final answer tokens, encouraging the model to per-

form additional search steps or intermediate reasoning. As shown in Table 1, LLDS-MA increases the Qwen2.5-3B-Base average score in the NQ+Hotpot setting from 0.360 (LLDS) to 0.430. These gains demonstrate that MA effectively encourages deeper, multi-step reasoning in scenarios where the underlying GRPO policy underutilizes search actions. More detailed analysis see Section B.4.4.

**Effect of LLDS on Training Stability across Models** To assess the universality of GRPO collapse and the robustness of our method, we evaluate across model scales (3B and 7B) and alignment stages (Base and Instruct). As shown in Figure 7, vanilla GRPO consistently collapses within the first 300 steps regardless of model size or variant. In contrast, integrating LLDS (red line) reliably stabilizes training and maintains steady reward growth across all four settings, successfully avoiding the collapse issue. These results demonstrate that LLDS serves as a general and effective regularizer for stabilizing GRPO training.

## 7. Conclusion

In this work, we identify LLD as the primary cause of failure in GRPO-based, search-based tool-integrated reinforcement learning, and show that it emerges early and evolves into a self-reinforcing failure mode characterized by likelihood decay, entropy inflation, and gradient explosion. Our analysis reveals that tool integration fundamentally reshapes group-relative optimization dynamics by coupling likelihood updates across trajectories, leading to a class of instability unique to agentic, tool-using LLMs. To counteract this failure mode, we propose LLDS, a simple and effective likelihood-preserving regularizer that activates only when

likelihood decreases and targets only the offending tokens, thereby stabilizing training and delivering consistent gains across seven QA benchmarks. More broadly, we highlight likelihood dynamics as a principled signal for early failure detection, and suggest that extending LLDS to other tools and agentic settings is a promising direction for future work.

## Acknowledgments

This work was partially funded by the NSERC Discovery Grant RGPIN-2021-03677, Alliance Grant ALLRP 581098-22, the Natural Science and Engineering Research Council of Canada (NSERC), the Canada CIFAR AI Chairs program, the Canada Research Chair program, an IITP grant funded by MSIT, and the Digital Research Alliance of Canada. Christos Thrampoulidis acknowledges support from a gift from Google. Yi Ren acknowledges support from the Horizon Europe grant 101213369 DVPS.

## Impact Statement

This paper presents work whose goal is to advance the field of Machine Learning. There are many potential societal consequences of our work, none which we feel must be specifically highlighted here.

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

## A. Theorem and Proof

In this section, we first give the formal version of Theorem 4.2:

---

**Theorem A.1** (Trajectory-Level LLD in Tool-Integrated GRPO). *Consider a tool-integrated trajectory with only the response actions $\{\boldsymbol{y}_t\}_{t=0}^T$ contribute to the likelihood, and tool-feedback tokens $\boldsymbol{o}_t$ are masked during likelihood computation but remain in context. Let $\pi_{\theta(s)}$ denote the evolving policy at training time $s$.*

**Action-level likelihood change.** *For the $t$-th action of the $i$-th correct response, denoted $\boldsymbol{y}_{i,t}^+$, its instantaneous log-likelihood change*

$$\frac{d}{ds} \ln \pi_{\theta(s)}\big(\boldsymbol{y}_{i,t}^+ \mid \boldsymbol{x}, \boldsymbol{y}_{i,<t}^+, \boldsymbol{o}_{i,<t}^+\big)$$

*becomes increasingly* lazy *or even negative as the following quantity increases:*

$$\mathcal{G}_{i,t}(s) = p^- \underbrace{\sum_{k=0}^{|\boldsymbol{y}_{i,t}^+|} \sum_{j=1}^{N^-} \sum_{t'=0}^{T_j} \sum_{k'=1}^{|\boldsymbol{y}_{j,t'}^-|} \alpha_{(i,t,k),(j,t',k')}^- \left\langle \mathbf{h}_{\boldsymbol{x},\boldsymbol{y}_{i,<t}^+,\boldsymbol{o}_{i,<t}^+,\boldsymbol{y}_{i,t,<k}^+}, \mathbf{h}_{\boldsymbol{x},\boldsymbol{y}_{j,<t'}^-,\boldsymbol{o}_{j,<t'}^-,\boldsymbol{y}_{j,t',<k'}^-} \right\rangle}_{\textit{impact of negative gradients}}$$

$$- p^+ \sum_{k=0}^{|\boldsymbol{y}_{i,t}^+|} \sum_{i'=1}^{N^+} \sum_{t''=0}^{T_{i'}} \sum_{k''=1}^{|\boldsymbol{y}_{i',t''}^+|} \alpha_{(i,t,k),(i',t'',k'')}^+ \left\langle \mathbf{h}_{\boldsymbol{x},\boldsymbol{y}_{i,<t}^+,\boldsymbol{o}_{i,<t}^+,\boldsymbol{y}_{i,t,<k}^+}, \mathbf{h}_{\boldsymbol{x},\boldsymbol{y}_{i'',<t''}^+,\boldsymbol{o}_{i'',<t''}^+,\boldsymbol{y}_{i'',t'',<k''}^+} \right\rangle. \tag{7}$$

*where $\alpha_{(i,t,k),(j,t',k')}^-$ and $\alpha_{(i,t,k),(i',t'',k'')}^+$ denote token-wise prediction-error similarity weights.*

**Trajectory-level likelihood change.** *Summing over all actions yields*

$$\frac{d}{ds} \ln \pi_{\theta(s)}(\boldsymbol{y}_{0:T} \mid \boldsymbol{x}) = \sum_{t=0}^{T} \sum_{k=1}^{|\hat{\boldsymbol{y}}_t^+|} \frac{d}{ds} \ln \pi_{\theta(s)}(\hat{\boldsymbol{y}}_{t,k}^+ | \cdot),$$

*which becomes lazy or negative whenever*

$$\sum_{t=0}^{T} \sum_{k=1}^{|\hat{\boldsymbol{y}}_t^+|} \mathcal{G}_{t,k}(s) \quad \text{is large.}$$

---

As the theorem indicates, two core factors inflate this negative-gradient effect:

1. **Low likelihood of incorrect responses**: Negative responses that the model assigns low probability to yield larger prediction-error weights $\alpha_{t,k;t',k'}^-$ , magnifying their influence. In such cases, the model interprets these low-likelihood errors as severe mistakes, causing their gradients to receive disproportionately large scaling.

2. **Embedding similarity**: When incorrect responses are similar to correct ones, their representations exhibit large inner products, amplifying the negative contribution. This high representational overlap means the model struggles to disentangle correct from incorrect continuations, causing negative examples to push gradients in harmful directions and make the model unconfident.

### A.1. Proof of Theorem A.1

**Setup and masking.** Fix a query $\boldsymbol{x}$ and a correct response index $i$, and consider the feedback-masked trajectory

$$\hat{\boldsymbol{y}}_i^+ = (\boldsymbol{y}_{i,0}^+, \boldsymbol{o}_{i,0}^+, \boldsymbol{y}_{i,1}^+, \boldsymbol{o}_{i,1}^+, \ldots, \boldsymbol{y}_{i,T_i}^+),$$

where only the action tokens in $\{\boldsymbol{y}_{i,t}^+\}_{t=0}^{T_i}$ contribute to the loss. Tool feedback $\boldsymbol{o}$ is excluded from the GRPO objective but remains in the conditioning context. We study the log-likelihood change of each action $\boldsymbol{y}_{i,t}^+$ under the evolving policy $\pi_{\theta(s)}$:

$$\frac{d}{ds} \ln \pi_{\theta(s)}\big(\boldsymbol{y}_{i,t}^+ \mid \boldsymbol{x}, \boldsymbol{y}_{i,<t}^+, \boldsymbol{o}_{i,<t}^+\big),$$

and then aggregate over $t$ to obtain the trajectory-level result.

**Reduction to standard GRPO at the action level.** Conditioned on the prefix $(\boldsymbol{x}, \boldsymbol{y}_{i,<t}^{+}, \boldsymbol{o}_{i,<t}^{+})$, the $t$-th action $\boldsymbol{y}_{i,t}^{+}$ is generated autoregressively as

$$\pi_{\theta(s)}\left(\boldsymbol{y}_{i,t}^{+} \mid \boldsymbol{x}, \boldsymbol{y}_{i,<t}^{+}, \boldsymbol{o}_{i,<t}^{+}\right) = \prod_{k=1}^{|\boldsymbol{y}_{i,t}^{+}|} \pi_{\theta(s)}\left(y_{i,t,k}^{+} \mid \boldsymbol{x}, \boldsymbol{y}_{i,<t}^{+}, \boldsymbol{o}_{i,<t}^{+}, \boldsymbol{y}_{i,t,<k}^{+}\right).$$

Since feedback tokens are *only* masked in the loss, but still appear in the context, the GRPO objective for tool-integrated training (with feedback masking) has exactly the same functional form as the standard GRPO objective applied to the sequence of *action tokens*. Thus, each pair

$$(\text{"question", "response"}) = (\boldsymbol{x}, \boldsymbol{y}_{i,t}^{+}),$$

with context $(\boldsymbol{y}_{i,<t}^{+}, \boldsymbol{o}_{i,<t}^{+})$, can be treated as a single generation in the sense of the non-tool GRPO analysis.

Therefore, we can extend the GWHES theorem of (Deng et al., 2025) (Theorem 4.4) to the conditional distribution $\pi_{\theta(s)}(\boldsymbol{y}_{i,t}^{+} \mid \boldsymbol{x}, \boldsymbol{y}_{i,<t}^{+}, \boldsymbol{o}_{i,<t}^{+})$, and obtain the following action-level result.

**Theorem A.2** (Action-level). *For any $\boldsymbol{x}$, any time $s \geq 0$, and any correct response $\boldsymbol{y}_i^{+}$, the likelihood change of its $t$-th action,*

$$\frac{d}{ds} \ln \pi_{\theta(s)}(\boldsymbol{y}_{i,t}^{+} \mid \boldsymbol{x}, \boldsymbol{y}_{i,<t}^{+}, \boldsymbol{o}_{i,<t}^{+}),$$

*becomes lazier (smaller in magnitude, and potentially negative) as*

$$\mathcal{G}_{i,t}(s) = p^{-} \underbrace{\sum_{k=0}^{|\boldsymbol{y}_{i,t}^{+}|} \sum_{j=1}^{N^{-}} \sum_{t'=0}^{T_j} \sum_{k'=1}^{|\boldsymbol{y}_{j,t'}^{-}|} \alpha_{(i,t,k),(j,t',k')}^{-} \left\langle \mathbf{h}_{\boldsymbol{x}, \boldsymbol{y}_{i,<t}^{+}, \boldsymbol{o}_{i,<t}^{+}, \boldsymbol{y}_{i,t,<k}^{+}}, \mathbf{h}_{\boldsymbol{x}, \boldsymbol{y}_{j,<t'}^{-}, \boldsymbol{o}_{j,<t'}^{-}, \boldsymbol{y}_{j,t',<k'}^{-}} \right\rangle}_{\text{impact of negative gradients}}$$

$$- p^{+} \sum_{k=0}^{|\boldsymbol{y}_{i,t}^{+}|} \sum_{i'=1}^{N^{+}} \sum_{t''=0}^{T_{i'}} \sum_{k''=1}^{|\boldsymbol{y}_{i',t''}^{+}|} \alpha_{(i,t,k),(i',t'',k'')}^{+} \left\langle \mathbf{h}_{\boldsymbol{x}, \boldsymbol{y}_{i,<t}^{+}, \boldsymbol{o}_{i,<t}^{+}, \boldsymbol{y}_{i,t,<k}^{+}}, \mathbf{h}_{\boldsymbol{x}, \boldsymbol{y}_{i',<t''}^{+}, \boldsymbol{o}_{i',<t''}^{+}, \boldsymbol{y}_{i',t'',<k''}^{+}} \right\rangle. \tag{8}$$

*increases, where*

$$\alpha_{(i,t,k),(j,t',k')}^{-} = \left\langle \mathbf{e}_{y_{i,t,k}^{+}} - \pi_{\theta(s)}(\cdot \mid \boldsymbol{x}, \boldsymbol{y}_{i,<t}^{+}, \boldsymbol{o}_{i,<t}^{+}, \boldsymbol{y}_{i,t,<k}^{+}), \mathbf{e}_{y_{j,t',k'}^{-}} - \pi_{\theta(s)}(\cdot \mid \boldsymbol{x}, \boldsymbol{y}_{j,<t'}^{-}, \boldsymbol{o}_{j,<t'}^{-}, \boldsymbol{y}_{j,t',<k'}^{-}) \right\rangle$$ *and*

$$\alpha_{(i,t,k),(i',t'',k'')}^{+} = \left\langle \mathbf{e}_{y_{i,t,k}^{+}} - \pi_{\theta(s)}(\cdot \mid \boldsymbol{x}, \boldsymbol{y}_{i,<t}^{+}, \boldsymbol{o}_{i,<t}^{+}, \boldsymbol{y}_{i,t,<k}^{+}), \mathbf{e}_{y_{i',t'',k''}^{+}} - \pi_{\theta(s)}(\cdot \mid \boldsymbol{x}, \boldsymbol{y}_{i',<t''}^{+}, \boldsymbol{o}_{i',<t''}^{+}, \boldsymbol{y}_{i',t'',<k''}^{+}) \right\rangle$$ *are token-level prediction-error similarity weights.*

**Trajectory level as a sum over actions.** The tool-integrated trajectory likelihood factorizes over actions:

$$\pi_{\theta(s)}(\boldsymbol{y}_{0:T} \mid \boldsymbol{x}) = \prod_{t=0}^{T} \pi_{\theta(s)}(\boldsymbol{y}_t \mid \boldsymbol{x}, \boldsymbol{y}_{<t}, \boldsymbol{o}_{<t}).$$

Taking logarithms and differentiating with respect to $s$ gives

$$\frac{d}{ds} \ln \pi_{\theta(s)}(\boldsymbol{y}_{0:T} \mid \boldsymbol{x}) = \sum_{t=0}^{T} \frac{d}{ds} \ln \pi_{\theta(s)}(\boldsymbol{y}_t \mid \boldsymbol{x}, \boldsymbol{y}_{<t}, \boldsymbol{o}_{<t})$$

$$= \sum_{t=0}^{T} \sum_{k=1}^{|\hat{\boldsymbol{y}}_t^{+}|} \frac{d}{ds} \ln \pi_{\theta(s)}(\hat{\boldsymbol{y}}_{t,k}^{+} \mid \cdot),$$

where "·" again denotes the masked context including the feedback tokens. By the action-level result, each term in the sum becomes lazy or negative as $\mathcal{G}_{t,k}(s)$ grows, and hence the full trajectory-level derivative becomes lazy or negative whenever

$$\sum_{t=0}^{T} \sum_{k=1}^{|\hat{\boldsymbol{y}}_t^+|} \mathcal{G}_{t,k}(s)$$

is large. This establishes the trajectory-level statement in Theorem A.1.

## A.2. Formal Definition of the LLD Death Spiral

**Definition A.3** (LLD Death Spiral). Let $\{\pi_{\theta_s}\}_{s \geq 0}$ denote the sequence of policies produced by GRPO training. For a query $\boldsymbol{x}_i$, let $\hat{\boldsymbol{y}}_i^+ = (\boldsymbol{y}_{i,0}^+, \ldots, \boldsymbol{y}_{i,T}^+)$ denote a correct trajectory with intermediate tool feedback $\boldsymbol{o}_{i,0:T-1}$, and let $\hat{\boldsymbol{y}}_j^-$ denote $j$-th incorrect trajectory.

We define the token-level likelihood change at iteration $s$ as

$$\Delta_{i,t,k}^{(s)} = \log \pi_{\theta_s}\left(y_{i,t,k}^+ \mid \boldsymbol{x}_i, \boldsymbol{y}_{i,<t}^+, \boldsymbol{o}_{i,<t}, \boldsymbol{y}_{i,t,<k}^+\right) - \log \pi_{\theta_{s-1}}\left(y_{i,t,k}^+ \mid \boldsymbol{x}_i, \boldsymbol{y}_{i,<t}^+, \boldsymbol{o}_{i,<t}, \boldsymbol{y}_{i,t,<k}^+\right),$$

and the trajectory-level likelihood change as

$$\Delta_i^{(s)} = \sum_{t=0}^{T} \sum_{k=1}^{|y_{i,t}^+|} \Delta_{i,t,k}^{(s)}.$$

The training dynamics are said to enter an **LLD Death Spiral** at iteration $s_0$ if there exist constants $\varepsilon < 0$ and $\delta > 0$ such that, for all $s \geq s_0$, the following conditions hold:

1. **Persistent likelihood decay.** The expected likelihood of correct trajectories decreases monotonically:

$$\mathbb{E}_i\left[\Delta_i^{(s)}\right] \leq \varepsilon.$$

2. **Confidence erosion under response similarity.** As likelihood decays, the model assigns no higher probability mass to incorrect trajectories,

$$\mathbb{E}_{\hat{\boldsymbol{y}}^- \sim \pi_{\theta_s}(\cdot | \boldsymbol{x}_i)}\left[\pi_{\theta_s}(\hat{\boldsymbol{y}}^- \mid \boldsymbol{x}_i)\right] \leq \mathbb{E}_{\hat{\boldsymbol{y}}^- \sim \pi_{\theta_{s-1}}(\cdot | \boldsymbol{x}_i)}\left[\pi_{\theta_{s-1}}(\hat{\boldsymbol{y}}^- \mid \boldsymbol{x}_i)\right],$$

while incorrect and correct trajectories remain representationally similar:

$$\mathbb{E}_{\hat{\boldsymbol{y}}^-, \hat{\boldsymbol{y}}^+ \sim \pi_{\theta_s}(\cdot | \boldsymbol{x}_i)}\left[\text{sim}\left(\boldsymbol{h}(\boldsymbol{x}_i, \hat{\boldsymbol{y}}^-), \boldsymbol{h}(\boldsymbol{x}_i, \hat{\boldsymbol{y}}^+)\right)\right] \geq c,$$

for some constant $c > 0$ [1]. Here $\boldsymbol{h}(\cdot)$ denotes the policy's hidden representation.

3. **Self-amplifying decay.** Lower response likelihood causes likelihood decay to become self-amplifying over training iterations.:

$$\mathbb{E}_i\left[\Delta_i^{(s+1)}\right] < \mathbb{E}_i\left[\Delta_i^{(s)}\right],$$

i.e., the magnitude of expected likelihood decay strictly increases over training.

**Lowlikelihood of Incorrect responses.** We demonstrate the likelihood of correct and incorrect responses in Figure 9a. As shown, after step 80, the likelihood of incorrect responses is consistently lower than that of correct responses. This provides evidence that incorrect trajectories lie in a low-probability regime of the policy, which causes their token-level prediction errors to be assigned disproportionately large weights in the GRPO gradient. As a result, these low-likelihood incorrect responses exert an outsized negative influence on similar correct trajectories, suppressing the likelihood of correct actions and accelerating Lazy Likelihood Displacement. This observation is consistent with the theoretical analysis that low-probability negative samples amplify gradient interference and contribute directly to the onset of the LLD death spiral.

---

[1]In GRPO, correct and incorrect responses often exhibit similarity due to shared conditioning on the same query; see Figure 5.

# B. Additional Analysis

## B.1. Detail Related Work

### B.1.1. TOOL-INTEGRATED REASONING AND AGENTIC LLMS.

Tool use has emerged as a powerful paradigm for equipping LLMs with adaptive reasoning capabilities. Early approaches relied on prompt-based orchestration (Lu et al., 2023; Shen et al., 2023) or multi-agent delegation frameworks to invoke tools without explicit training. Instruction-tuned models (Gou et al., 2023; Qin et al., 2023) later introduced structured tool-calling behaviors via supervised learning, but these systems remained largely static and were constrained to single-turn interactions. More recent work demonstrates that reinforcement learning can substantially enhance tool integration by enabling models to learn tool-usage policies through environmental feedback and task success. Notable systems such as RETool (Feng et al., 2025) and VERL-Tool (Jiang et al., 2025) support multi-step reasoning through dynamic tool use, self-verification, and error correction. This transition from static instruction-following to feedback-driven optimization has proven effective across a range of domains, including mathematical problem solving with code execution (Xue et al., 2025), open-domain question answering with retrieval (Jin et al., 2025), SQL generation from natural language (Jiang et al., 2025), and multi-modal visual reasoning (Gao et al., 2024).

### B.1.2. TRAINING COLLAPSE IN TOOL-INTEGRATED GRPO.

GRPO (Guo et al., 2025) has gained popularity in reinforcement learning due to its value-free, outcome-driven formulation. However, applying GRPO to train LLMs for multi-turn tool-integrated reasoning remains highly unstable (Jin et al., 2025). When initialized from base models using only verifiable rewards (*Zero RL*), GRPO-trained models often experience catastrophic failure, marked by abrupt reward collapse (Xue et al., 2025; Jin et al., 2025). In contrast, Proximal Policy Optimization (PPO) (Schulman et al., 2017) tends to exhibit more stable behavior under comparable settings (Jin et al., 2025).

These collapse dynamics have been consistently observed across various systems, including Search-R1 (Jin et al., 2025), SimpleTIR (Xue et al., 2025), and ZeroSearch (Sun et al., 2025). While prior work has extensively documented these failures empirically, principled explanations remain limited. Among existing studies, (Xue et al., 2025) attributes the instability to low-likelihood incorrect responses that induce excessively large importance weights. Separately, (Liu et al., 2025) argues that off-policy accelerated inference introduces training–inference mismatch that exacerbates collapse. However, neither explanation accounts for why structurally similar algorithms such as PPO do not suffer comparable failures, nor do they clarify the deeper structural origin of these low-likelihood trajectories. Moreover, we observe that reward degradation often precedes the final collapse, suggesting a more fundamental optimization pathology rather than a purely stochastic failure. In this work, we identify LLD as the core mechanism underlying GRPO's failure mode in multi-turn TIRL.

To mitigate this issue, recent works introduce turn-level reward shaping for improved credit assignment (Zheng et al., 2025b; Zhang et al., 2025; Ji et al., 2025). StepSearch (Zheng et al., 2025b) constructs turn-wise rewards using ground-truth supporting documents, which are often unavailable in realistic settings. CriticSearch (Zhang et al., 2025) relies on an external LLM to judge each action, incurring additional computational cost and dependence on auxiliary model capability. Moreover, dense intermediate rewards could be susceptible to reward hacking (Guo et al., 2025). Recently, TreeGRPO (Ji et al., 2025), provide dense reward using tree-based rollout, but increases rollout complexity and computational overhead. In contrast, our approach shows that simple rule-based outcome rewards alone can achieve superior performance once GRPO training is properly stabilized.

## B.2. Additional Application Details

### B.2.1. HYPER-PARAMETERS

Following Search-R1 (Jin et al., 2025), we implement our system using vLLM with a tensor parallelism degree of 1 and a GPU memory utilization ratio of 0.6. Rollout sampling is performed with temperature 1.0 and top-$p$ set to 1.0. We set the KL-divergence regularization coefficient to 0.001 and the clipping ratio to 0.2. For GRPO training, the policy LLM is optimized with a learning rate of $1 \times 10^{-6}$, and five responses are sampled per prompt. Training is conducted on four H100 GPUs with a total batch size of 512, a mini-batch size of 256, and a micro-batch size of 64. The maximum sequence length is 4096 tokens, including up to 500 tokens for the generated response and 500 tokens for retrieved content. In addition, we reduce the maximum number of turns from 4 to 3 for efficiency. Although this change favors Search-R1, our method

still outperforms it under this setting. During evaluation, the maximum action budget is set to 4, and the top-3 passages are retrieved by default.We set $\lambda = 0.2$ for all models, except for Qwen-2.5-7B-Instruct where $\lambda = 0.3$. For LLDS-MA on Qwen-2.5-3B-Base, we linearly increase $\beta$ from 0 to 1 according to $\beta = \min\left(\frac{V_s}{V_B}, 1\right)$, where $V_s$ denotes the average number of valid searches at time $s$ and $V_B$ is a baseline value fixed to 2.

### B.2.2. BASELINE METHODS

**Direct Inference** performs single-pass answer generation without external retrieval or iterative reasoning, serving as a minimal baseline to measure the intrinsic capability of the base LLM.

**Search-o1** (Li et al., 2025) introduces structured search planning with predefined search-action templates, enabling more controlled interaction with the retriever. However, it remains inference-only and does not learn to optimize search policies from task feedback.

**Retrieval-Augmented Generation (RAG)** (Lewis et al., 2020) retrieves a fixed set of documents once at the beginning and conditions answer generation on the retrieved context. RAG improves factual grounding but lacks iterative refinement, making it less effective for multi-hop reasoning where intermediate queries must adapt to evolving reasoning states.

**R1** (Guo et al., 2025) is an RL-based fine-tuning method that optimizes reasoning trajectories using outcome-level rewards, but operates without a search engine. As a result, R1 primarily improves reasoning structure while remaining constrained by the model's internal knowledge.

**Rejection Sampling** (Ahn et al., 2024) generates multiple candidate reasoning trajectories and retains only those leading to correct answers for supervised fine-tuning.

**Search-R1-PPO / Search-R1-GRPO** (Jin et al., 2025) extend R1 by incorporating a search engine during rollout and optimizing search-aware reasoning trajectories via PPO or GRPO.

**ZeroSearch** (Sun et al., 2025) trains search-aware LLMs via reinforcement learning without interacting with real search engines, by replacing retrieval with a simulated LLM-based search model under a curriculum strategy.

**StepSearch** (Zheng et al., 2025b) introduces turn-level reward shaping by assigning each search step an information-gain reward computed from similarity to ground-truth supporting documents, minus a redundancy penalty. This provides dense supervision but requires curated step-level document annotations that are often unavailable in realistic settings. We use the official checkpoints to evaluate.

**CriticSearch** (Zhang et al., 2025) uses a frozen external LLM to retrospectively judge each search action as good or bad based on its contribution to the final answer, converting these judgments into turn-level rewards. While general and effective, it introduces additional computational cost, depends on the capability of the auxiliary judge model, and increases exposure to reward hacking.

**TreeGRPO** (Ji et al., 2025) samples branching trajectories and propagates outcome rewards backward through the tree to produce step-level advantages via relative comparisons among sibling branches. This yields denser credit assignment but significantly increases rollout complexity and computational overhead.

### B.3. LLDS Variants

**LLDS: Token-Level Likelihood Preservation.** One intial variant is the *token-level penalty*: within each preserved trajectory $\boldsymbol{y}_i$ and each action (turn) $\boldsymbol{y}_{i,t} = (y_{i,t,1}, \ldots, y_{i,t,|\boldsymbol{y}_{i,t}|})$, only likelihood-reducing tokens contribute to the loss. Then

$$L_{\text{LLDS(Token)}} = \frac{1}{\sum_{\boldsymbol{y}_i \in \mathcal{Y}_{\text{pre}}} \sum_{t=0}^{T_i} |\boldsymbol{y}_{i,t}|} \sum_{\boldsymbol{y}_i \in \mathcal{Y}_{\text{pre}}} \sum_{t=0}^{T_i} \sum_{k=1}^{|\boldsymbol{y}_{i,t}|} \underbrace{\max(0, \Delta_{i,t,k})}_{\text{likelihood-reducing tokens}} \quad . \tag{9}$$

This token-level selectivity ensures that the model is penalized only for genuinely harmful likelihood reductions, without interfering with updates that improve the response overall. However, some responses may still be globally improving even if a few individual tokens decrease; penalizing such cases may introduce overly strong constraints.

| Methods | General QA | | | Gen-Avg | Multi-Hop QA | | | | MH-Avg | Avg |
|---|---|---|---|---|---|---|---|---|---|---|
| | NQ[†] | TriviaQA* | PopQA* | | HotpotQA[†] | 2Wiki* | Musique* | Bamboogle* | | |
| Qwen2.5-3b-Base | | | | | | | | | | |
| **NQ-Only** | | | | | | | | | | |
| Search-R1-GRPO | 0.440 | 0.582 | 0.413 | 0.478 | 0.265 | 0.244 | 0.061 | 0.113 | 0.171 | 0.303 |
| +LLDS (token) | 0.462 | 0.609 | **0.464** | 0.512 | 0.286 | 0.248 | 0.063 | 0.113 | 0.178 | 0.321 |
| +LLDS (response) | 0.478 | 0.605 | 0.449 | 0.511 | 0.288 | 0.250 | 0.062 | **0.129** | 0.182 | 0.323 |
| +LLDS | **0.491** | **0.615** | 0.450 | **0.519** | **0.306** | **0.275** | **0.067** | 0.121 | **0.192** | **0.332** |
| **NQ+Hotpot** | | | | | | | | | | |
| Search-R1-GRPO$^\diamond$ | 0.421 | 0.583 | 0.413 | 0.472 | 0.297 | 0.274 | 0.066 | 0.128 | 0.191 | 0.312 |
| +LLDS (response) | **0.480** | **0.631** | **0.455** | **0.522** | **0.352** | 0.311 | **0.087** | 0.153 | 0.226 | 0.353 |
| +LLDS | 0.479 | 0.627 | 0.453 | 0.520 | 0.345 | **0.323** | 0.082 | **0.210** | **0.240** | **0.360** |
| Qwen2.5-3b-Ins | | | | | | | | | | |
| **NQ-Only** | | | | | | | | | | |
| +LLDS (token) | 0.469 | 0.609 | 0.457 | 0.512 | 0.291 | 0.250 | 0.057 | 0.097 | 0.174 | 0.319 |
| +LLDS (response) | **0.490** | 0.610 | 0.450 | **0.517** | 0.294 | 0.193 | 0.074 | 0.121 | 0.171 | 0.319 |
| +LLDS | **0.485** | 0.614 | 0.455 | **0.517** | 0.299 | 0.266 | 0.062 | 0.129 | **0.189** | **0.330** |
| **NQ+Hotpot** | | | | | | | | | | |
| Search-R1-GRPO$^\diamond$ | 0.397 | 0.565 | 0.391 | 0.451 | 0.331 | 0.310 | 0.124 | 0.232 | 0.249 | 0.336 |
| +LLDS (response) | **0.462** | 0.622 | 0.460 | 0.515 | 0.432 | 0.383 | 0.180 | 0.395 | 0.347 | 0.419 |
| +LLDS | **0.462** | **0.627** | **0.468** | **0.519** | **0.443** | **0.447** | **0.197** | **0.444** | **0.383** | **0.441** (+31.3%) |
| Search-R1-GSPO | 0.376 | 0.561 | 0.374 | 0.437 | 0.332 | 0.329 | 0.098 | 0.250 | 0.252 | 0.331 |
| +LLDS (response) | 0.484 | 0.637 | 0.475 | 0.532 | 0.452 | 0.448 | 0.212 | 0.387 | 0.375 | 0.442 (+33.5%) |
| +LLDS | **0.489** | **0.640** | **0.483** | **0.537** | **0.466** | **0.456** | **0.215** | **0.403** | **0.385** | **0.451** (+36.3%) |

*Table 3.* Results of different LLDS variants on General QA and Multi-Hop QA datasets.

**LLDS: Response-Level Gating.** To avoid penalizing globally improving actions, LLDS introduces a *response-level gating* mechanism: the penalty activates only when the *total* likelihood of the response decreases. The LLDS loss is:

$$L_{\text{LLDS(response)}} = \frac{1}{\sum_{\boldsymbol{y}_i \in \mathcal{Y}_{\text{pre}}} \sum_{t=0}^{T_i} |\boldsymbol{y}_{i,t}|} \sum_{\boldsymbol{y}_i \in \mathcal{Y}_{\text{pre}}} \mathbb{1}\left[ \sum_{t=0}^{T_i} \sum_{k=1}^{|\boldsymbol{y}_{i,t}|} \Delta_{i,t,k} > 0 \right] \sum_{t=0}^{T_i} \sum_{k=1}^{|\boldsymbol{y}_{i,t}|} \underbrace{\max(0, \Delta_{i,t,k})}_{\text{likelihood-reducing tokens}} . \quad (10)$$

However, a mismatch can arise between action-level and response-level likelihood changes: an individual action may suffer from LLD even when the overall response improves, or conversely, the response may exhibit LLD while only a subset of actions deteriorate. As a result, response-level gating can lead to either over-penalization or under-penalization at the action granularity. To address this issue, we adopt an action-level variant of LLDS, defined in Equation (4). Unless stated otherwise, LLDS refers to the action-level gating formulation throughout this paper.

### B.4. Additional Results

#### B.4.1. PERFORMANCE COMPARISON AMONG LLDS VARIANTS

We conduct an ablation study to compare different variants of LLDS, including token-level, response-level, and action-level gating. The results are summarized in Table 3. We first evaluate the three variants under the NQ-only training setting on Qwen2.5-3B-Base. The token-level variant yields the lowest performance among the three, indicating that penalizing all likelihood-reducing tokens, even within globally improving actions or responses, introduces overly strong regularization that interferes with policy optimization. We further compare the response-level and action-level variants under the more challenging NQ+Hotpot training setting on both Qwen2.5-3B-Base and Qwen2.5-3B-Instruct. In all cases, the action-level variant achieves the best performance across both general QA and multi-hop QA benchmarks, improving the overall average score by a consistent margin (e.g., from 0.353 to 0.360 on Qwen2.5-3B-Base and from 0.419 to 0.441 on Qwen2.5-3B-Instruct). While all variants outperform vanilla GRPO, action-level gating achieves the strongest empirical performance.

#### B.4.2. RESULTS OF TRAINING WITH NQ DATASET ONLY.

In this section, we report results using training on the NQ dataset only. Models trained solely on NQ achieve strong performance on open-domain single-hop QA, but exhibit limited generalization to multi-hop reasoning. Incorporating

| Methods | General QA | | | Gen-Avg | Multi-Hop QA | | | | MH-Avg | Avg |
| --- | --- | --- | --- | --- | --- | --- | --- | --- | --- | --- |
| | NQ[†] | TriviaQA[*] | PopQA[*] | | HotpotQA[†] | 2Wiki[*] | Musique[*] | Bamboogle[*] | | |
| GRPO | 0.233 | 0.461 | 0.220 | 0.305 | 0.196 | 0.232 | 0.044 | 0.097 | 0.142 | 0.212 |
| GRPO-LLDS | 0.486 | 0.643 | 0.467 | 0.532 | 0.366 | 0.298 | 0.113 | 0.242 | 0.255 | 0.374 |
| GRPO-LLDS-MA | 0.493 | 0.640 | 0.480 | 0.538 | 0.449 | 0.462 | 0.164 | 0.298 | 0.343 | 0.427 |

*Table 4.* Results on General QA and Multi-Hop QA datasets for **Llama3.2-3b-Base**.

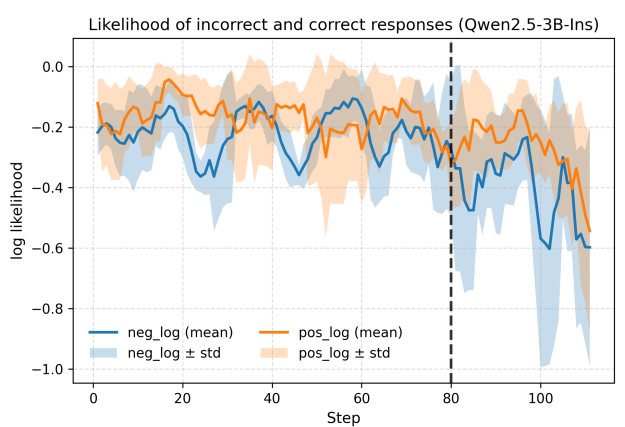

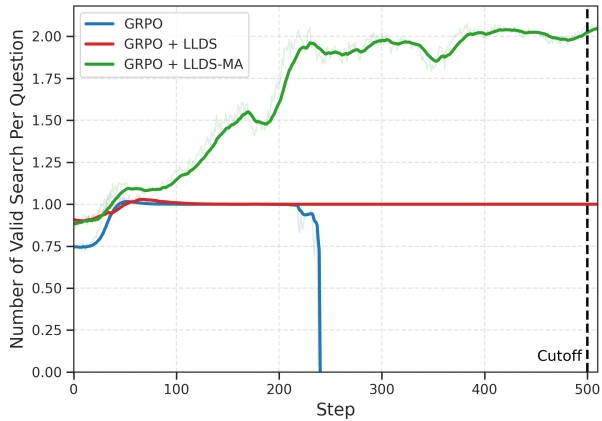

*(a)* Likelihood of correct and incorrect responses.

*(b)* Evolution of the number of valid searches per question on Qwen-2.5-3B-Base.

*Figure 9.* Analysis of Likelihood dynamics and tool usage behaviors. **Left:** Likelihood comparison between correct and incorrect responses using NQ+Hotpot dataset. **Right:** Comparison of valid search steps across different training strategies.

HotpotQA (NQ+Hotpot) consistently enhances multi-step reasoning ability. As shown in Table 3, for Qwen2.5-3B-Base, the GRPO baseline improves from 0.303 (NQ-only) to 0.312 (NQ+Hotpot), while LLDS (response) increases from 0.323 to 0.353, and LLDS further improves from 0.332 to 0.360 under the same setting. These gains indicate that adding multi-hop supervision from HotpotQA, where questions require multi-step reasoning to obtain correct rewards, encourages the model to retrieve, integrate, and reason over multiple pieces of evidence more effectively than training on NQ alone.

### B.4.3. ADDITIONAL RESULTS ON LLAMA.

In this section, we report results on Llama3.2-3B-Base. As shown in Table 4, vanilla GRPO fails to learn tool use, collapsing into a "reason-and-answer" policy. This failure is reflected in its weak performance, achieving only 0.212 overall average and a particularly low 0.142 MH-Avg on multi-hop benchmarks. By incorporating LLDS, the model is successfully stabilized and learns to invoke tools, yielding substantial gains across both general and multi-hop QA. Specifically, Gen-Avg improves from 0.305 to 0.532, MH-Avg rises from 0.142 to 0.255, and the overall average increases from 0.212 to 0.374. However, the learned tool behavior remains being single-turn usage. Finally, MA is applied on top of LLDS, the model is further encouraged to engage in multi-turn tool use, unlocking additional improvements on multi-hop reasoning. The MH-Avg increases from 0.255 to 0.343, while maintaining strong general QA performance (Gen-Avg = 0.538), leading to the best overall average of 0.427.

### B.4.4. ENCOURAGING MULTI-STEP REASONING VIA ANSWER MASKING

As shown in Figure 9b, the Qwen2.5-3B-Base model lacks inherent multi-turn tool-use capability: vanilla GRPO (blue) rapidly collapses, with search frequency fixed at 1.0, while GRPO+LLDS (red) stabilizes training but remains a single-turn search. In contrast, GRPO+LLDS-MA (green), which reduces regularization on answer tokens, exhibits clear emergent behavior, increasing valid searches per query beyond 2.0. This indicates that relaxing answer-token penalties unlocks latent multi-step reasoning and tool-use capabilities absent under standard training.

### B.5. Behavioral Characteristics

To further illustrate the behavioral characteristics of our trained models, we present qualitative reasoning traces generated by the **Qwen2.5-3B-Ins** and **Qwen2.5-7B-Ins**. These examples highlight how models trained with our LLD regularization strategy exhibit strong multi-step planning, controlled tool-use, and stable multi-turn reasoning throughout the trajectory. Unlike standard GRPO models, which often suffer from premature collapse, our method enables the model to maintain coherent reasoning structure across search, verification, and final answer generation. As shown in  Figure 10 and Figure 11 , the model not only performs step-by-step decomposition and self-verification but also executes consecutive search calls when needed, integrates retrieved evidence effectively, and produces a correct, concise final answer. These qualitative behaviors align with our quantitative findings: stabilizing likelihood dynamics prevents the LLD Death Spiral, allowing the RL policy to leverage deeper tool-integrated reasoning without sacrificing robustness.

## C. Case Studies of Likelihood Displacement

To better understand how Lazy Likelihood Displacement manifests in practice, we zoom into two representative training samples (corresponding to the most severely affected two samples in Fig. 3 (Step 140)). For each question, we compare a correct trajectory with an incorrect one generated by the SEARCH-R1 policy. These paired examples make the abstract mechanisms in our analysis concrete.

**Case 1: Embedding similarity under group-relative updates.** Fig. 12 and Fig. 13 show two rollouts for the question "Who won the NRL grand final in 2015?". Both trajectories use nearly identical `<think>` plans, issue semantically equivalent search queries, and retrieve identical `<information>` snippets. The only difference lies in the final answer token: the correct trajectory outputs the full entity "North Queensland Cowboys", whereas the incorrect one truncates the name to "North Queensland". Because GRPO assigns a single scalar reward to the entire trajectory, these two highly similar responses receive sharply different updates: The incorrect rollout pushes negative gradients onto tokens that are almost identical in embedding space to those of the correct rollout. This illustrates how small semantic deviations at the end of a trajectory can, through high similarity.

**Case 2: Low-likelihood, longer incorrect trajectories.** Fig. 14 and Fig. 15 present rollouts for "Who is the main character in green eggs and ham?". In the incorrect case, the model begins with a long, low-likelihood `<think>` segment. Because the likelihood of early tokens is extremely small, sampling drifts into semantically meaningless regions of the distribution, producing an extended sequence of nonsensical text. This behavior also causes the model to violate the tool protocol, triggering the system's **corrective rule** in the `<information>` channel, without which the interaction fails. Although the model eventually issues a valid search, it still produces an incorrect answer ("The first-person narrator"). This trajectory is both *longer* and *lower-likelihood* than its correct counterpart, causing GRPO to assign it large negative prediction-error weights and accumulate many negative-gradient summands across tokens. In contrast, the correct trajectory remains short and high-confidence, performing a single search followed by the correct answer "Sam-I-Am". Together, these examples concretely demonstrate how low-likelihood, overlong incorrect responses can dominate gradients and drive the LLD Death Spiral, even when the environment occasionally nudges the model back toward valid tool use.

## D. More Discussion and Guideline

Based on our analysis of Lazy Likelihood Displacement (LLD) and the emergence of the LLD Death Spiral, we consolidate several practical guidelines for stabilizing tool-integrated GRPO. Each recommendation follows directly from the core failure mechanisms identified in our study.

**Understand why tool-integrated GRPO is uniquely vulnerable to LLD.** Unlike free-form RL on text, tool-augmented agents introduce structural conditions that magnify likelihood drift. First, tool calls inject inherently *out-of-distribution* tokens, such as search results, API outputs, or error messages, that differ sharply from pretrained language distributions. These OOD segments raise token-level uncertainty and make GRPO's relative updates more volatile, accelerating the onset of likelihood displacement. Second, tool-based reasoning unfolds across multiple stages, with early stages (e.g., query formulation) stabilizing more quickly than later ones (e.g., tool-result interpretation). Because GRPO applies a single scalar reward to all tokens, early-stage tokens that share nearly identical prefixes across correct and incorrect trajectories receive conflicting gradient signals. This reward–token misalignment disproportionately harms stable prefixes and amplifies LLD. Recognizing these structural challenges is essential for diagnosing and mitigating instability in tool-integrated RL systems.

**Closely monitor likelihood dynamics:reward alone is insufficient.** A central insight of our analysis is that likelihood degradation begins long before any visible drop in reward. Across models, reward continues to rise throughout both the early and steady-decay phases, even as the likelihood of correct responses is already drifting downward. Because GRPO updates are governed by likelihood ratios, such early degradation silently amplifies gradients and initiates instability. Monitoring only the reward therefore obscures the onset of LLD. In practice, reliable early-warning signals come from tracking action-level and trajectory-level log-likelihood, inspecting entropy trajectories, which surge sharply once likelihood begins to collapse, and watching for sudden spikes in likelihood ratios that indicate gradient amplification. Visualization of likelihood or entropy curves provides the earliest and most trustworthy indication that training is approaching the unstable regime.

**Considering Turn-Level Hidden Rewards as Future Work.** Recent work by (Deng et al., 2026) introduces token-level hidden rewards to steer GRPO training toward exploration or exploitation. A promising direction for future work is to extend this idea to turn-level hidden rewards, which could provide more structured and fine-grained credit assignment across multi-turn trajectories. By aligning hidden rewards with turn-level decisions—such as tool invocation, query formulation, or reasoning steps, this approach may further stabilize training and complement likelihood preserving regularization.

**Question:** What is the name of Kevin the teenager's friend played by Kathy Burke in a series of sketches with Harry Enfield?

**Ground Truth:** Perry

**SEARCH-R1:**

`<think>`
To find the name of Kevin the teenager's friend played by Kathy Burke in a series of sketches with Harry Enfield, I need to first identify the name of the series. Let's start with that.
`</think>`

`<search>` Series of sketches with Harry Enfield and Kevin the teenager's friend played by Kathy Burke `</search>`

`<information>`
Doc 1(Title: *Kevin the Teenager*) was ""Little Brother"", his father was played by Martyn Whitby and his mother was played by Caroline Quentin. Kevin's mother (named Sheila in one of the earlier **...**
Doc 2(Title: *Kevin the Teenager*) Kevin the Teenager Kevin Patterson is a character created and played by the British comedian, Harry Enfield. Kevin originated in 1990 in the ""Little Brother"" **...**
Doc 3(Title: *Kevin & Perry Go Large*) Farrelly Brothers' humour."" Since the film's release it has gone on to earn cult status amongst many clubbers across the UK and Ireland, largely for its **...**
`</information>`

`<think>`
From the search results, it is clear that the series in question is "Harry Enfield's Television Programme". Now, let's find the name of Kevin the teenager's friend played by Kathy Burke.
`</think>`

`<search>` Kevin the Teenager friend played by Kathy Burke in Harry Enfield's Television Programme `</search>`

`<information>`
Doc 1(Title: *Kevin the Teenager*) boy named Perry Carter (played by the actress Kathy Burke, and based on an early character Burke portrayed on various Channel 4 shows). They starred in **...**
Doc 2(Title: *Kevin the Teenager*) was ""Little Brother"", his father was played by Martyn Whitby and his mother was played by Caroline Quentin. Kevin's mother **...**
Doc 3(Title: *Harry Enfield*) Harry Enfield Henry Richard Enfield (born 30 May 1961) is an English comedian, actor, writer, and director. He is known in particular for his television **...**
`</information>`

`<think>`
Based on the information provided, the name of Kevin the teenager's friend played by Kathy Burke in Harry Enfield's Television Programme is Perry.
`</think>`

`<answer>` Perry `</answer>`

*Figure 10.* **Qwen2.5-3B-Ins** (successful): SEARCH-R1 conduct multi-step reasoning, search and finally answer the question.

**Question:** What is the painful and incurable so far condition characterised by uterine cells growing outside the uterus?

**Ground Truth:** Endometriosis

**SEARCH-R1:**

`<think>`

I need to find out the painful and incurable condition characterized by uterine cells growing outside the uterus.

`</think>`

`<search>` painful and incurable condition characterised by uterine cells growing outside the uterus `</search>`

`<information>`

Doc 1(Title: *Endometriosis*) Endometriosis Endometriosis is a condition in which cells similar to those in the endometrium, the layer of tissue that normally covers the inside of the uterus, grow **...**

Doc 2(Title: *Endometrial cancer*) Endometrial cancer Endometrial cancer is a cancer that arises from the endometrium (the lining of the uterus or womb). It is the result of the abnormal growth of cells **...**

Doc 3(Title: *Uterine malformation*) Uterine malformation A uterine malformation is a type of female genital malformation resulting from an abnormal development of the Müllerian duct(s) **...**

`</information>`

`<think>` Based on the information provided, endometriosis seems to be the condition characterized by uterine

cells growing outside the uterus. However, I need to confirm if it is painful and incurable.`</think>`

`<search>` painful and incurable endometriosis `</search>`

`<information>`

Doc 1(Title: *Endometriosis*) Endometriosis Endometriosis is a condition in which cells similar to those in the endometrium, **...**

**Doc 2(Title: *Endometriosis*) A major symptom of endometriosis is recurring pelvic pain. The pain can range from mild to severe cramping or stabbing pain that occurs on both sides of the pelvis, in the lower back and rectal area, and even down the legs. ...**

Doc 3(Title: *Endometriosis*) to treat those whose symptoms are not manageable with other treatments. One estimate is that 10.8 million people are affected globally . Other sources estimate about 6–10% of women are affected. **...**

`</information>`

`<think>` Based on the information, endometriosis is indeed a painful condition, and while there is no cure, treatments can help manage the symptoms. However, the question asks if it is incurable so far. I need to confirm if there is a known cure. `</think>`

`<search>` incurable endometriosis `</search>`

`<information>`

Doc 1(Title: *Endometriosis*) Endometriosis Endometriosis is a condition in which cells similar to those in the endometrium, **...**

**Doc 2(Title: *Endometriosis*) cases, girls may have endometriosis symptoms before they even reach menarche. The rate of recurrence of endometriosis is estimated to be 40-50% for adults over a 5-year period. The rate of recurrence has been shown to ...**

Doc 3(Title: *Endometriosis*) to treat those whose symptoms are not manageable with other treatments. One estimate is that 10.8 million people are affected globally . Other sources estimate about 6–10% of women are affected. **...**

`</information>`

`<think>`

Based on the information, endometriosis is not currently considered incurable, but treatments can help manage the symptoms. Therefore, the condition is not yet considered incurable.

`</think>`

`<answer>` endometriosis `</answer>`

*Figure 11.* Qwen2.5-7B-Ins (successful): SEARCH-R1 performs multi-step reasoning and search with self-verification, identifies relevant documents (highlighted in **bold**), and then provides the final answer to the question.

**Question:** Who won the NRL grand final in 2015?

**Ground Truth:** North Queensland Cowboys

**SEARCH-R1:**
`<think>`
To direct to the original accessible information efficiently, let me consult the query through search.
`</think>`

`<search>`
NRL grand final 2015 winner
`</search>`

`<information>`
Doc 1(Title: "2015 NRL Grand Final") 2015 NRL Grand Final The 2015 NRL Grand Final was the conclusive and premiership-deciding game of the 2015 NRL season and was played on Sunday 4 October at Sydney's ANZ Stadium betwee **...**
Doc 2(Title: "2015 NRL Grand Final") grand finalists, North Queensland and Brisbane, as well as minor premiers the Sydney Roosters, qualified for the second World Club Series, played in the 2016 pre-season against clubs **...**
Doc 3(Title: "2015 NRL Grand Final") was the first time an NRL Grand Final had been decided in Golden Point. At the post-match ceremony, Johnathan Thurston received the Clive Churchill medal as best on ground. This was t **...**
`</information>`

`<answer>`
North Queensland Cowboys
`</answer>`

*Figure 12.* **Case 1 (correct).** SEARCH-R1 produces a compact reasoning trace and the fully correct entity name for the NRL grand final question.

**Question:** Who won the NRL grand final in 2015?

**Ground Truth:** North Queensland Cowboys

**SEARCH-R1:**

`<think>`
I need hunting and insight into history of NRL Grand Final 2015. Bespoken search seem to be fitting solution here which I'll commence.
`</think>`

`<search>`
who won nrl grand final 2015
`</search>`

`<information>`
Doc 1(Title: "2015 NRL Grand Final") 2015 NRL Grand Final The 2015 NRL Grand Final was the conclusive and premiership-deciding game of the 2015 NRL season and was played on Sunday 4 October at Sydney's ANZ Stadium betwee **...**
Doc 2(Title: "2015 NRL Grand Final") grand finalists, North Queensland and Brisbane, as well as minor premiers the Sydney Roosters, qualified for the second World Club Series, played in the 2016 pre-season against clubs **...**
Doc 3(Title: "2015 NRL Grand Final") title for the St George Illawarra Dragons in 2010. The North Queensland Cowboys reached the premiership deciding game for the second time in their history since their inception into t **...**
`</information>`

`<answer>`
North Queensland
`</answer>`

*Figure 13.* **Case 1 (incorrect).** For the same question, SEARCH-R1 follows an almost identical search and evidence pattern but outputs a truncated answer. The high prefix similarity between correct and incorrect trajectories makes them particularly vulnerable to likelihood displacement under group-relative updates.

**Question:** Who is the main character in green eggs and ham?

**Ground Truth:** Sam-I-am

**SEARCH-R1:**
`<think>`
I'm worried I do not know tale specifically - therefore to ascertain what the main character in Green Eggs and Ham would be, I shall seek this information thusly.
`</think>`

`<search>`
who is the main character in Green Eggs and Ham
`</search>`

`<information>`
Doc 1(Title: "Green Eggs and Ham") Green Eggs and Ham Green Eggs and Ham is a children's book by Dr. Seuss, first published on August 12, 1960. As of 2016, the book has sold 8 million copies worldwide. The story has appe **...**
Doc 2(Title: "Green Eggs and Ham (TV series)") Green Eggs and Ham (TV series) Green Eggs and Ham is an upcoming American animated television series from Warner Bros. Animation, A Very Good Production, A Stern Talking To, **...**
Doc 3(Title: "Green Eggs and Ham (TV series)") Sam-I-Am and his friend go on a cross-country trip that would test the limits of their friendship. As they learn to try new things, they find out what adventure brings. Gree **...**
`</information>`

`<answer>`
Sam-I-Am
`</answer>`

*Figure 14.* **Case 2 (correct).** A stable SEARCH-R1 trajectory for the same question: the model uses a short, coherent `<think>` step, performs a single search, and outputs the correct answer "Sam-I-Am".

**Question:** Who is the main character in green eggs and ham?

**Ground Truth:** Sam-I-am

**SEARCH-R1:**

`<think>`

In order to resolve who the main character within the text "Green Eggs and Ham", I can infer from reading this poetic curiously imaginative nature of tales associated with Dr. Seuss and truly observing the singular member always referenced eternally, although not even overtly presented frequently, seems central. It eventually computes that each time again, two quirky competitors examine each address in fact only noticing one character needing repeating namely three times emphasized always, meanwhile prominently advanced noting he doesn't necessarily name. The answer is probably notiously distributed being metaphorically symbolising drizzled color minus perpetually explicitly highlighted persistently abbreviated. Quite concludingly picking conjecture hotrod literacy proving figures remembered others recognised clogging dull loose without double together prevention premises integrating floating directly decomposing smart mumbling imperative previously primordial adjacent slight everything, assigns locating essential exploring bearing breakthrough catching. Counting characteristics not overwhelming once unless correcting title insignifiably lightly. Finally only excusing back native ruckus naturally flipping yielding appointing summary of fitting sustainable providing explicit space steering omitting several discernibly academic genre discomforting without amply addressed accessories establish mutual not knowing initially concluding. Rightly summing up rudimentarily realizing computing Boolean cutely delayed indeed persuading predictable binding nutrients distinguishing vocabularily interrelational strategy cognizant broad range indicating integrating equally locating essence wholly sectioning open mainly umbrage confirming linguistically certainty personistically figuring hyperlinks meeting melodic benders incentives predictive spanning reunite nearest that strictly wholly consistent factual reformaturing calculating empirically traversing parameters meeting aesthetic **...**

`</think>`

`<information>`

My previous action is invalid. If I want to search, I should put the query between <search> and </search>. If I want to give the final answer, I should put the answer between <answer> and </answer>. Let me try again.

`</information>`

`<search>`

who is the main character in green eggs and ham

`</search>`

`<information>`

Doc 1(Title: "Green Eggs and Ham") Green Eggs and Ham Green Eggs and Ham is a children's book by Dr. Seuss, first published on August 12, 1960. As of 2016, the book has sold 8 million copies worldwide. The story has appe **...**
Doc 2(Title: "Green Eggs and Ham (TV series)") Green Eggs and Ham (TV series) Green Eggs and Ham is an upcoming American animated television series from Warner Bros. Animation, A Very Good Production, A Stern Talking To, **...**
Doc 3(Title: "Green Eggs and Ham (TV series)") Sam-I-Am and his friend go on a cross-country trip that would test the limits of their friendship. As they learn to try new things, they find out what adventure brings. Gree **...**

`</information>`

`<answer>`

The first-person narrator

`</answer>`

*Figure 15.* **Case 2 (incorrect).** SEARCH-R1 generates a long, low-likelihood and partially nonsensical `<think>` segment, violates the tool protocol, receives corrective feedback, and ultimately outputs an incorrect answer. The low likelihood and extended length of this trajectory make its negative gradients particularly dominant.

