# OpenReview forum: "On Group Relative Policy Optimization Collapse in Agent Search: The Lazy Likelihood-Displacement"
_ICML.cc/2026/Conference — ICML 2026 regular_

### Official Review · Reviewer_cvQ4 · 2026-03-05

**Soundness:** 3
**Presentation:** 3
**Significance:** 3
**Originality:** 3
**Overall Recommendation:** 5
**Confidence:** 4

**Summary:**

The authors investigate the degradation of a model's confidence in a correct response during training using the Lazy Likelihood Displacion Problem (GRPO) method. They consider the problem for Tool-Integrated GRPO, emphasizing that in this training setup, degradation is amplified, leading to early training collapse. To prevent this, the authors propose introducing an additional component into the training loss, penalizing the degradation of confidence in the generations from the previous step with a positive advantage. The method provides measurements on QA-based and retrieval-based benchmarks and is compared with GRPO (in Search-R1 mode) and GSPO.

**Compliance With Llm Reviewing Policy:**

Affirmed.

**Final Justification:**

The paper presents a method for preventing the degradation of a model's confidence in the correct generation during training using the GRPO method with tool calling and subsequent training process collapse. The authors identify and substantiate the problem, finding empirical support for it, and propose a modification to the training objective by adding a component that directly addresses the problem. The authors conduct a wide range of evaluations on various datasets and compare them with a large number of baselines. During the rebuttal, the authors addressed all my concerns and conducted additional evaluations using broader set of benchmarks and model architectures further confirming the effectiveness of the method. Therefore, I am adjusting my score to "accept".

**Key Questions For Authors:**

1. It would be good to hear from the authors more of the motivation behind the chosen confidence enhancement formula. Why, for example, instead of LLDS, didn't the authors choose a KL enhancement between the log-probability distributions for the correct answer from the policy from the previous step and the policy from the current step?

2. It would be useful to hear how exactly the authors determine the set of positive examples N+. Is this done for the entire question dataset at each step, or is a subset of similar question-positive answer pairs sampled?

3. It would be useful to see the dynamics of the model's average confidence in the correct answer during training with the proposed method, similar to how the authors provided for the GRPO models in Figures 2, 3, and 5.

4. It would be great if the authors provided a visualization of the dynamics of the model's response entropy (both for all answers and for the best-worst ones) during training with the method.

5. As a result of GRPO training, the entropy of the model's responses often collapses (responses become more deterministic). However, in Figure 4, the authors observe the opposite behavior. It would be nice to hear clarification on this point.

6. It would be useful to see the method's performance on datasets from more diverse domains—tool integrated math (e.g., GSM-8k, AIME-24/25 with TiR), coding (SWE Bench, SWE Rebench, etc.)—to understand the method's portability to other domains.

7. It would be helpful if the authors provided measurements of the method on other architectures (llama/gemma/deepseek/...).

8. It would be especially interesting to understand whether the method is transferable to MoE architectures. I would appreciate the authors' comments on this matter.

**Limitations:**

yes

**Strengths And Weaknesses:**

**Strengths:**
- The authors provide empirical and theoretical justifications for LLD, showing graphs of how the recall, log probabilities, and entropy progress depending on the training step.
- The authors conduct studies of the method on models of different sizes and demonstrate the transferability of their method to various Tool Integrated Reasoning RL methods: GRPO and GSPO.
- The authors study the influence of the lambda hyperparameter (the weight of the LLD loss addition) on the training process.
- The authors explore the generally relevant problem of model degradation in reasoning-style RL, which is relevant for agent learning.

**Weaknesses:**
- Lack of evidence of transferability to other models. The authors conduct studies on Qwen models, without exploring models from other families (gemma, llama, deepseek).
- The authors investigate problems related to tool calling for retrieval and quality assurance. No convincing arguments are provided indicating the existence of a similar problem in other domains. For example, when solving mathematical problems with access to the Python tool or coding tasks

---

> ### Author Rebuttal · Authors · 2026-03-29
>
> Thank you for your evaluation and suggestions. We truly appreciate your time and address your comments point by point below.
>
> **Supp link**: https://anonymous.4open.science/r/ID6198_Rebuttal_Supplementary-1574/ID6198_Rebuttal_Supp.pdf
>
> ----- W1 & Q7 -----
>
> **Answer:** Thank you for this question. We respectfully disagree that our study is limited to Qwen models. In Appendix Section B.4.3 and Tab. 4, we also report results on Llama3.2-3B-Base, where we observe the same pattern: vanilla GRPO collapses, while LLDS stabilizes training and substantially improves performance. Specifically, on Llama3.2-3B-Base, the overall average improves from 0.212 to 0.427, and the multi-hop average improves from 0.142 to 0.343.
>
> ----- W2 & Q6 -----
>
> **Answer:** As reflected in our title, our paper focuses on agent search, so we study QA under the Search-R1 setup, which is a strong and widely used benchmark for tool-integrated RL and therefore an appropriate testbed for analyzing collapse.
>
> More broadly, we believe the mechanism is not specific to search, but to multi-turn tool-integrated RL. While this paper focuses on agent search, following your suggestion, we also study a kernel code generation task [1] using Qwen3-8B. For efficiency, we use a setup with at most 2 turns and a maximum response length of 4000 tokens per turn; the compiler serves as the tool, and compilation errors and feedback are inserted into the context as tool feedback, after which the model is asked to revise the code to fix errors and improve performance. As shown in Fig. 3 in Supp. link, vanilla GRPO starts dropping at around 60 steps, while GRPO+LLDS stabilizes training and leads to clear performance gains, as shown below:
>
> | Method            | Correctness | Fast\@1 | Fast\@1.2 | Fast\@1.5 |
> |-|-|-|-|-|
> | GRPO              | 0.356      | 0.244  | 0.133    | 0.019    |
> | GRPO+LLDS (Ours)  | **0.598**  | **0.452** | **0.168** | **0.084** |
>
> These results support that our findings are not restricted to QA, but extend to other practically important multi-turn tool-use domains as well.
>
> [1] KernelBench: Can LLMs Write Efficient GPU Kernels?
>
> ----- Q1 -----
>
> **Answer:** A KL term would regularize the entire token distribution across all turns, including tokens and turns that are not responsible for LLD. This over-constrains optimization and does not distinguish between good updates and harmful likelihood decay. In contrast, our LLDS design is more surgical: it minimizes disruption to training and is activated only when a turn’s likelihood decreases, and only on the specific offending tokens.  In multi-turn tool-integrated settings, this action-level awareness is important because different turns may evolve in opposite directions.
>
> ----- Q2 -----
>
> **Answer:** Thank you for your question, $N^+$ is determined within each training batch, rather than over the entire dataset. Specifically, for a given prompt, we sample a group of responses and identify the positive examples using the reward.
>
> ----- Q3 & 4 -----
>
> **Answer:** Thank you for this valuable suggestion. Following your suggestion, we additionally plot the likelihood (which inversely reflects entropy) trajectory of correct responses. As shown in Fig. 1 in the Supp. link, LLDS stabilizes the likelihood of correct responses, while vanilla GRPO exhibits severe LLD. We will include this in the revised version.
>
>
> ----- Q5 -----
>
> **Answer:** Thank you for this question. In standard single-turn RL without tool-use, entropy often decreases as the model becomes more deterministic. However, we observe the opposite in multi-turn tool-integrated RL: entropy increases sharply before collapse as shown in Fig. 4. This reflects the distinct failure regime from the usual entropy collapse.  In our paper, we find two factors that drive the model toward uncertainty rather than over-confidence:
>
> 1)  tool feedback introduces OOD information, pushing responses into a lower-likelihood regime and amplifying the LLD effect;
>
> 2) under outcome-based rewards, a turn in an overall incorrect trajectory can still be correct and highly similar to a turn in a correct trajectory.
>
> As a result, the model does not simply become more deterministic but can instead become increasingly uncertain, with rising entropy preceding collapse.
>
> ----- Q8 -----
>
> **Answer:** We believe LLDS should be transferable to MoE architectures, since the method is model-agnostic and does not rely on any architecture-specific modification. LLDS operates purely at the level of policy likelihood dynamics during RL training, by selectively regularizing likelihood suppression on preserving trajectories, and is therefore not tied to a particular backbone design. We’ve shown the transferability across different model families in Sec. B.4.1. We thank the reviewer for raising this interesting question and agree that an explicit MoE evaluation would be valuable future work.

---

> > ### Author Rebuttal · Reviewer_cvQ4 · 2026-04-01
> >
> > I thank the authors for the additional experimental results and explanations. Nevertheless, some of the authors' claims remain unconvincing to me.
> >
> > ---
> > **Question**: It would be especially interesting to understand whether the method is transferable to MoE architectures. I would appreciate the authors' comments on this matter.
> >
> > **Response**: We believe LLDS should be transferable to MoE architectures, since the method is model-agnostic and does not rely on any architecture-specific modification. LLDS operates purely at the level of policy likelihood dynamics during RL training, by selectively regularizing likelihood suppression on preserving trajectories, and is therefore not tied to a particular backbone design. We've shown the transferability across different model families in Sec. B.4.1. We thank the reviewer for raising this interesting question and agree that an explicit MoE evaluation would be valuable future work.
> >
> > The core of my concern is that RL training dynamics in MoE architectures differ fundamentally from those in dense models, primarily due to routing instability. In particular, routing can directly affect the very mechanism of embedding similarity between correct and incorrect responses within a generation group — similar token spans across different responses may be processed by different experts, altering their hidden representations in ways that the paper's theoretical analysis does not account for. Additionally, by penalizing likelihood decreases, LLDS may inadvertently penalize the router network itself, which could negatively impact training stability and convergence.
> >
> > Overall, the convergence of LLDS in MoE settings remains an open question for me. It would be considerably more convincing if the authors could provide experimental results on at least one MoE architecture.
> >
> > ---
> > **Question**: It would be useful to hear how exactly the authors determine the set of positive examples N+. Is this done for the entire question dataset at each step, or is a subset of similar question-positive answer pairs sampled?
> > **Response**: Thank you for your question, N+ is determined within each training batch, rather than over the entire dataset. Specifically, for a given prompt, we sample a group of responses and identify the positive examples using the reward.
> >
> > Could the authors provide a more detailed explanation here? Specifically: what happens when none of the sampled generations for a given prompt leads to a correct answer — is this prompt skipped, and if so, does this introduce any bias? Furthermore, have the authors investigated how the variance in the number of correct responses per prompt within a batch affects convergence?
> >
> >  ---

---

> > > ### Author Response · Authors · 2026-04-05
> > >
> > > We are glad that our responses have addressed part of your questions, and we resolve the remaining two point by point below.
> > >
> > > ------- Q1 -------
> > >
> > > **Answer**:  Thank you for this insightful question, which broadens the scope of our work. While routing may introduce some variability, existing evidence [1]  indicates that the same token processed by different experts does not result in wildly different token embeddings (last layer output embedding), thus our theoretical analysis remains valid for MoE.
> > >
> > >
> > > Due to computational constrains, we use Phi-mini-MoE-instruct[2] model, which has 7.6B total parameters and 2.4B activated. Due to computational constraints and the model’s 4096-token context limit, we use a reduced setting with top-k=2 retrieved chunks, maximum chunk length 400, maximum total response length 1500 and maximum turns =2.
> > >
> > > First, we still observe a clear and consistent LLD effect in the MoE model:
> > >
> > > | pos_logp | 1 | 5 | 10 | 15 | 20 | 25 | 30 | 35 | 40 |
> > > |---|---:|---:|---:|---:|---:|---:|---:|---:|---:|
> > > |GRPO| -7.74|  -7.72 | -7.72 | -7.78 | -7.89 | -9.13 |  -9.84 |  -12.09  | -16.67 |
> > > |GRPO+LLDS| -7.58|  -7.60 | -7.61 | -7.51 | -7.40 | -7.13 |  -7.14 |  -6.89  | -6.82  |
> > >
> > > Due to time constraints, we train for 100 steps and report the results below. We observe that vanilla GRPO collapses, while LLDS show improved performance:
> > >
> > > | Metric | nq | triviaqa | popqa | hotpotqa | 2wikimultihopqa | musique | bamboogle | mean |
> > > |---|---:|---:|---:|---:|---:|---:|---:|---:|
> > > |GRPO (best ckpt) | 0.211| 0.358 | 0.218 | 0.113 | 0.101 | 0.043 | 0.184 | 0.175 |
> > > |GRPO (50 ckpt) | 0.0| 0.0 | 0.0 | 0.0 | 0.0 | 0.0 | 0.0 | 0.0 |
> > > |GRPO+LLDS | 0.259 | 0.454 | 0.288 | 0.181 | 0.193 |0.061 | 0.232 | **0.238** |
> > >
> > > These preliminary results suggest that the LLD phenomenon persists in MoE models, and that LLDS show potential in stabilizing MoE.
> > >
> > > [1] A Closer Look into Mixture-of-Experts in Large Language Models
> > >
> > > [2] SlimMoE: Structured Compression of Large MoE Models via Expert Slimming and Distillation.
> > >
> > > ---- Q2 -------
> > >
> > > > what happens when none of the sampled generations for a given prompt leads to a correct answer — is this prompt skipped, and if so, does this introduce any bias?
> > >
> > > **Answer**: Thank you for this question. We follow the standard GRPO training pipeline and do not filter or skip prompts when none of the sampled generations is correct. Therefore, our method does not introduce additional bias from prompt selection.
> > >
> > > >  Furthermore, have the authors investigated how the variance in the number of correct responses per prompt within a batch affects convergence?
> > >
> > > **Answer**:  Thank you for your question. Our method is robust to the correctness ratio (i.e., the number of correct responses per prompt). Since we use binary outcome rewards (0/1), the **reward curve directly reflects the average correctness ratio per prompt across the batch**.
> > >
> > > **Robust to different initial correctness ratios.** As shown in Figure 7, LLDS consistently stabilizes training across settings with very different starting correctness levels. For example, Qwen-2.5-3B-Base starts below 10%, Qwen-2.5-3B-Instruct starts around 0.2, and Qwen-2.5-7B-Instruct starts around 0.3. Despite these differences, LLDS consistently prevents collapse and moves training toward a higher-performance region.
> > >
> > > **Robust throughout training.** LLDS also remains effective as the correctness ratio changes over the course of training. As reward increases, LLDS continues to stabilize optimization and support convergence to better final performance.
> > >
> > > These results suggest that LLDS is **robust both to the initial correctness ratio and to its variation during training**.
> > >
> > > Thank you for your evaluation and suggestions. We truly appreciate your time! We would appreciate if you would consider adjusting your score or let us know if there are any remaining questions we could address.

---

### Official Review · Reviewer_qpDt · 2026-03-10

**Soundness:** 2
**Presentation:** 3
**Significance:** 2
**Originality:** 2
**Overall Recommendation:** 4
**Confidence:** 3

**Summary:**

This work identifies the Lazy Likelihood-Displacement phenomenon in multi-turn tool-use LLM agent. It analyzes the failure reason of the RL training, and proposes a regularization term to mitigate this issue.

**Compliance With Llm Reviewing Policy:**

Affirmed.

**Final Justification:**

Thw additional explanation and results address my concerns. I am lean to accepting the paper currently.

**Key Questions For Authors:**

1. How does the LLD suppresion regularization prevent false positives, as it may also affect the tokens that should reduce the likelihood.

2. How LLD in multi-turn is different from single-turn setting? Do they share something in common or are different in nature.

**Limitations:**

Limitation section is not found in the current version.

**Strengths And Weaknesses:**

Strengths

1. The paper is well-written, with both empirical  and theoretical analysis

2. The experiment baselines are comprehensive.

3. The improvement seems significant.

Weaknesses

1. The contribution  seems limited to me. Given that Deng et al., 2025 identifies LLD in single-turn setting, this work seems a direct extension of it to multi-turn setting, which makes the contribution incremental.

2. The scope is also limited, given that it mainly identifies some failure mode in RL training, and propose some regularization to fix it.

3. I also doubt about the LLD suppresion regularization term, which is used to penalize the tokens with likelihood-reduction, but how does it prevent false positive?

4. For experiment part, the empirical study is mainly conducted in QA datasets, with 3B-scale models, so it is not clear if this issue persists with other tasks such as software-engineering, and large LLMs.

---

> ### Author Rebuttal · Authors · 2026-03-29
>
> Thank you for your evaluation and suggestions. We truly appreciate your time!
>
> **Supp link** : https://anonymous.4open.science/r/ID6198_Rebuttal_Supplementary-1574/ID6198_Rebuttal_Supp.pdf
>
> ---- W1&Q2 ----
>
> **Answer**: Deng et al. study single-turn, tool-free text generation. Our work addresses multi-turn, tool-integrated RL (TIRL), a setting that introduces fundamentally different challenges at the phenomenon, analysis, and method levels.
>
> **Different phenomenon**. In Deng et al., LLD appears as a mild per-sample inefficiency and training remains stable. In contrast, in TIRL we observe that LLD escalates into catastrophic collapse where training fails across all settings (Fig. 7). This collapse is the central problem we address and is not observed in prior single-turn work.
>
> **New technical challenges**. TIRL introduces OOD tool feedbacks that are masked from the loss but remain in the conditioning context, breaking the assumption that all tokens are in-distribution in prior analysis. Extending the analysis to trajectory-level likelihood under masked feedback requires re-deriving GRPO gradient dynamics across multiple actions (Thm. A.1), which is a non-trivial contribution specific to this setting.
>
> **Multi-turn TIRL is fundamentally different**. As discussed in Appendix lines 979–987, multi-turn TIRL introduces two key effects absent in single-turn non-tool settings:
> 1) OOD tool feedback → low-likelihood regime (Fig. 6), amplifying LLD;
> 2) Turn-level similarity, where some turns of incorrect trajectories overlap with correct ones (Fig. 5; supp. link Fig. 4), causing stronger interference.
>
> Together, these make LLD turn-dependent and significantly amplified, rather than a mild effect.
>
> **Prior work does not directly transfer.** Deng et al. treats the response holistically. In TIRL, different turns behave differently: some remain healthy, while others cause the harmful interference. We therefore propose turn-level gating to target problematic turns specifically. Empirically, LLDS stabilizes training across all settings where vanilla GRPO collapses, leading to substantial performance gains.
>
> In summary, this is not a direct extension of prior single-turn LLD work, but a study of a distinct multi-turn failure mode in TIRL, together with a new analysis and targeted solution.
>
> ----- W2 ------
>
> **Answer**: We’d like to emphasize the broad view and impact of our work. As noted in the related work (lines 55–67), multi-turn training collapse has been consistently observed across multiple works, and training stability is a fundamental bottleneck. Therefore, identifying and understanding this failure mode is itself an important and broadly relevant problem.
>
> Moreover, our contribution generalizes across different models and training settings.
> - Model sizes: 3B and 7B (Tabs. 1–2).
> - Model families: Qwen and Llama (Tab. 4, Appendix B.4.3).
> - Alignment stages: Base and Instruct variants (Tabs. 1–2, Fig. 7).
> - RL objectives: GRPO and GSPO (Tab. 1, bottom rows).
>
> To further demonstrate the significance of our findings, we show the collapse and LLDS’s fix in multi-turn kernel generation (Fig. 3 and Tab. 1 in the Supp. link), suggesting that the issue is not task-specific.
>
> We therefore believe this work offers a general stabilization principle that may be broadly useful for future agent training.
>
> ----- W3&Q1 -----
>
> **Answer**: To address your concerns on tokens whose likelihood should decrease, we clarify that LLDS is designed to address this through its hierarchical gating design at three levels
> so that useful behaviors are protected without disrupting normal policy learning.
> - **Response level**: applied only to preserving responses with non-negative advantage.
> - **Action level**: triggered only when an action’s total likelihood decreases, avoiding penalties on healthy turns within imperfect trajectories.
> - **Token level**: once activated, it regularizes only the tokens whose likelihood decreases, rather than all tokens uniformly.
>
> With this design, empirically, Fig. 8 shows that across different regularization strengths, the reward curves remain close to the training dynamics of vanilla GRPO before collapse, suggesting that LLDS does not over-regularize, while Tabs. 1 and 2 show about 30% average improvement over vanilla GRPO, allowing training to continue into higher-performance regions.
>
> ----- W4 -----
>
> **Answer**:  As reflected in the title, our paper focuses on agent search and studies QA under the Search-R1 setup, a widely used benchmark. Beyond 3B models, Tab. 2 also includes Qwen-2.5-7B in both Base and Instruct, where performance improves from 0.35 to 0.48 (+37.1%) for 7B-Base and from 0.396 to 0.483 (+22.0%) for 7B-Ins. We also show transfer to other settings, including GSPO and Llama (App. Tab. 4).
>
> Following your suggestion, we include a kernel generation task; **please refer to our response to W2 of Reviewer cvQ4 for details**. These results show our findings extend beyond QA to other tool-use domains.

---

> > ### Author Rebuttal · Reviewer_qpDt · 2026-04-02
> >
> > I appreciate the author to provide additional results. Some of my concerns have been addressed. But for the extention to multi-turn setting, I still do not think this is fundamentally different from single-turn setting, given that they may have distinct phenomenon, which is likely due to the tool-use and non tool-use scenarios. However, if for single-turn, LLD exists, for multi-turn case, the error will be accumulated and propagated, I think this is not surprising, therefore, given the previous work has identified this phenomenon, I think the contribution is incremental. Furthermore, although empirically validated of the proposed regularization term, it is still unclear WHY this term works, it lacks some rational explanation. For this reason, I lean to maintain the score or slightly increase to 3.5 if existed.

---

> > > ### Author Response · Authors · 2026-04-04
> > >
> > > We are glad that our responses have addressed part of your questions, and you are already willing to increase the score.
> > >
> > > >I still do not think this is fundamentally different from single-turn setting, given that they may have distinct phenomena, which is likely due to the tool-use and non tool-use scenarios. However, if for single-turn, LLD exists, for multi-turn case, the error will be accumulated and propagated, I think this is not surprising, therefore, given the previous work has identified this phenomenon
> > >
> > > **Answer**: We believe there may be a key misunderstanding in the framing. Our contribution is not simply “extending single-turn LLD to multi-turn.” The central distinction is **tool-free  (Deng et al.)  vs. tool-integrated RL (ours)**. Note that **tool-integrated RL (TIRL) is naturally multi-turn**: even a single tool call already induces a multi-step decision process (tool invocation → feedback → final answer). In that sense, we are glad the reviewer agrees that the distinct phenomenon, catastrophic GRPO collapse in TIRL, is tied to tool use.
> > > That said, we respectfully disagree that this is an unsurprising effect:
> > > -  A priori, it is **not obvious that LLD would be linked to collapse in TIRL**, nor that errors would simply accumulate in a way that leads to catastrophic instability. That said, prior work (Deng et al) didn’t link LLD to TIRL collapse.
> > > - While this may seem intuitive in hindsight, it was **not clear to us beforehand that this would be the key mechanism**.
> > >
> > > We also hope the reviewer recognizes the value of our algorithmic contribution, which leads to very clear and substantial performance gains.
> > >
> > > > empirically validated of the proposed regularization term, it is still unclear WHY this term works
> > >
> > > **Answer**: Thank you for your question. We take this chance to further give a detailed explanation from both a mechanistic and a theoretical perspective.
> > >
> > > At a high level, RL aims to increase the likelihood of correct responses and suppress incorrect ones. The issue we identify is that under GRPO, even correct actions can lose likelihood (LLD), leading to instability.
> > >
> > > **Mechanism (how LLDS works).** LLDS is a simple and targeted correction:
> > > - Detect: It checks whether the likelihood of a correct action decreases after an update.
> > > - Regularize: Only when such a decrease occurs, it applies a penalty proportional to the likelihood reduction of the affected tokens.
> > >
> > > The key insight is that LLDS is conditional rather than blanket regularization. It does not constrain all updates or all tokens, but intervenes only when the harmful effect occurs, leaving beneficial updates untouched. This allows it to stabilize training without hindering learning.
> > >
> > > **Theory (why this mechanism is correct).** LLDS is not ad hoc, it is directly motivated by our analysis. Theorem A.1 shows that likelihood reduction arises from two factors:
> > > - OOD tool feedback → low-likelihood regime (Fig. 6), amplifying LLD;
> > > - Turn-level similarity, where some turns of incorrect trajectories overlap with correct ones (Fig. 5; supp. link Fig. 4), causing stronger interference.
> > >
> > > These effects drive LLD into a self-reinforcing death spiral. LLDS is designed as a minimal intervention to break this loop by preventing harmful likelihood reduction at the turn level, thereby avoiding the low-likelihood regime. Consistent with this mechanism, Fig. 1 in the Supp. link shows that LLDS stabilizes the likelihood of correct responses, whereas vanilla GRPO exhibits severe LLD.
> > >
> > > We respectively hope the reviewer can consider adjusting your score or let us know if there are any remaining questions we could address.

---

### Official Review · Reviewer_5BTJ · 2026-03-11

**Soundness:** 3
**Presentation:** 3
**Significance:** 3
**Originality:** 3
**Overall Recommendation:** 4
**Confidence:** 4

**Summary:**

This paper identifies Lazy Likelihood Displacement (LLD) as the core cause of training collapse in GRPO for tool-integrated reinforcement learning, and proposes a lightweight regularization method LLDS (LLD Suppression), which achieves a maximum relative performance improvement of 45.2% across seven question answering (QA) benchmarks while stabilizing the training process.

**Compliance With Llm Reviewing Policy:**

Affirmed.

**Final Justification:**

My concerns have been adequately addressed.

**Key Questions For Authors:**

- The primary goal of LLDS is to prevent training collapse from massive gradients by smoothing or constraining them. Does this also weaken the model's ability to learn difficult but correct paths? I am interested in evaluating the model's performance training on more difficult training sets.
- Why does this collapse occur "lazy" after the reward starts to improve, rather than at the beginning of training? Are there any specific quantitative metrics to monitor this?
- Regarding embedding similarity, which specific tokens cause this? Are there any visualizations or semantic statistical analyses available?
- Although this paper focuses on tool-integrated search task, can this method mitigate collapse in general reasoning tasks? Is there a fundamental difference between the two in terms of the collapse mechanism?

**Limitations:**

Yes

**Strengths And Weaknesses:**

> Strengths

- The problem of RL training collapse is very important.
- The explanation of the LLD phenomenon is very intuitive; although it may not be theoretically optimal, the LLDS method is reasonable.
- The structure of the paper is reasonable.
- The phenomenon of collapse is solved across models of different sizes.

> Weaknesses

- LLDS forcibly requires that the likelihood of some tokens cannot decrease. Although this stabilizes training, it also introduces bias in the optimization objective, which may sacrifice the model's convergence performance.
- The task scenarios are relatively limited.

---

> ### Author Rebuttal · Authors · 2026-03-29
>
> Thank you for your evaluation and suggestions. We truly appreciate your time and address your comments point by point below.
>
> **Supp link** : https://anonymous.4open.science/r/ID6198_Rebuttal_Supplementary-1574/ID6198_Rebuttal_Supp.pdf
>
> ----- W1 -----
>
> **Answer:**  Thank you for your question. A key design goal of LLDS is to introduce minimal interference, and it therefore uses selective activation at three levels:
> - **Response level**: applied only to preserving responses with non-negative advantage.
> - **Action level**: triggered only when an action’s total likelihood decreases, avoiding penalties on healthy turns.
> - **Token level**: once activated, it regularizes only the tokens whose likelihood decreases, rather than all tokens uniformly.
>
> This makes LLDS a targeted correction, not a blanket bias against policy updates. Empirically, Fig. 8 shows that across different regularization strengths, the reward curves stay close to vanilla GRPO before collapse, suggesting no clear over-regularization. Moreover, Fig.7 shows that LLDS reaches a higher reward and better training stability, while Tabs. 1 and 2 show ~ 30% average improvement over vanilla GRPO. These results suggest that LLDS does not sacrifice convergence; rather, by preventing collapse, it allows training to continue into higher-performance regions that vanilla GRPO cannot reach.
>
> ----- W2 -----
>
> **Answer:** Thank you for your question. As reflected in the title, our paper focuses on agent search. Within this scope, we evaluate seven widely used QA tasks under the Search-R1 setup. More broadly, we show consistent gains across model sizes (3B/7B), model families (Qwen/Llama), and objectives (GRPO/GSPO), suggesting the benefit is not limited to a narrow task setup.
>
> To address your concern, we additionally include a kernel generation task, which is quite different from search-integrated QA. As shown in Fig. 3 in the Supp. link, vanilla GRPO shows the same collapse phenomenon, whereas LLDS stabilizes training; this stabilization also translates into improved downstream performance, as reported below:
>
> | Method            | Correctness | Fast\@1 | Fast\@1.2 | Fast\@1.5 |
> |-|-|-|-|-|
> | GRPO              | 0.356      | 0.244  | 0.133    | 0.019    |
> | GRPO+LLDS (Ours)  | **0.598**  | **0.452** | **0.168** | **0.084** |
>
> ----- Q1 -----
>
> **Answer:** Thank you for this question. Tab. 3 should have addressed your concern. We conduct an ablation comparing training on NQ versus NQ + the more challenging HotpotQA. As shown there, after adding the harder HotpotQA training data, performance on Qwen2.5-3B-Instruct improves from 0.33 to 0.44. The gains are especially notable on challenging multi-hop benchmarks, where the average multi-hop performance rises from 0.189 to 0.383. These results suggest that LLDS won’t hinder the learning of difficult trajectories.
>
> ----- Q2 -----
>
> **Answer:** Thank you for this question. The collapse appears “lazy” because its two main drivers, 1) similarity between correct and incorrect trajectories and 2) low response likelihood, are not strong at the beginning and take time to accumulate. Early in training, similarity is still relatively low and response likelihood is not yet very small, so GRPO updates remain relatively stable and reward can continue to improve. As training progresses, however, these factors strengthen: Fig. 5 and Supp. link Fig. 4 show that similarity between correct and incorrect trajectories increases over time, while Fig. 2 and Fig. 9(a) show that response likelihood decays over time. To monitor this,  as discussed in Appendix lines 990–997, response likelihood and entropy are useful metrics.
>
> ----- Q3 -----
>
> **Answer:** We thank the reviewer for this question. In Eq. 8, similarity is computed over the sum of all token embeddings within an action, rather than at the level of any single token. Our point is therefore not that one specific token causes LLD, but that the overall embedding similarity between incorrect and correct responses increases during training. Following the reviewer’s suggestion, we further average token embeddings within each response and compute the wrong–correct response similarity. As shown in Fig. 4 of the Supp. link, this similarity increases over training, especially for before-feedback actions, which is consistent with the collapse behavior observed in Figs. 3 and 5 of the main paper.
>
>
> ---- Q4 ----
>
> **Answer:** Since LLDS operates on likelihood changes at the token and action level, it can regularize likelihood-reducing updates in any GRPO setting. We believe its mechanism can generalize to general reasoning tasks. The fundamental difference is that tool integration amplifies LLD through (1) OOD tool feedback that pushes responses into lower-likelihood regimes (Fig. 6), and (2) correct turns appearing within incorrect trajectories, intensifying gradient interference (Fig. 5). These factors make the collapse severe and more consistent in tool-integrated settings.

---

> > ### Author Rebuttal · Reviewer_5BTJ · 2026-04-02
> >
> > Thank you for the detailed reply and additional explanations. My concerns have been adequately addressed.

---

> > > ### Author Response · Authors · 2026-04-03
> > >
> > > Thank you again for your positive feedback and for recognizing our contributions. We are very glad to know that our responses have addressed your concerns. We would appreciate if you would consider adjusting your score or let us know if there are any remaining questions we could address. We also thank you for your valuable suggestions, which will help us further enhance the paper.

---

### Official Review · Reviewer_qJTK · 2026-03-12

**Soundness:** 2
**Presentation:** 3
**Significance:** 3
**Originality:** 2
**Overall Recommendation:** 4
**Confidence:** 4

**Summary:**

This paper investigates the training collapse issue in GRPO-based reinforcement learning for tool-integrated LLM agents. The authors identify a phenomenon termed Lazy Likelihood Displacement (LLD): during the training process, the likelihood of correct responses continuously declines even as the reward steadily increases. The authors argue that this phenomenon triggers a "death spiral": the declining likelihood amplifies the gradients of incorrect trajectories, exerting a greater negative impact on the shared tokens between correct and incorrect trajectories, which ultimately leads to training collapse.

To mitigate this issue, the paper proposes LLD Suppression (LLDS), a regularization method designed to penalize the decrease in likelihood. Experiments conducted on search-integrated QA tasks demonstrate that LLDS successfully stabilizes the training process and achieves impressive performance gains.

**Compliance With Llm Reviewing Policy:**

Affirmed.

**Final Justification:**

This paper studies an important and practically relevant issue in RL training for tool-integrated LLM agents, namely training collapse in GRPO. I find the empirical observation of the pre-collapse dynamics (likelihood decay, entropy increase, and eventual reward collapse) valuable, and the proposed LLDS method is simple, easy to implement, and demonstrates clear empirical gains. These aspects support the paper’s significance and practical usefulness.

In summary, I view this work as a solid empirical contribution with practical impact and a useful mitigation strategy, but with limitations in mechanistic understanding and novelty. I maintain a weak accept recommendation.

**Key Questions For Authors:**

1. The paper claims that LLD is the core mechanism driving collapse. Can the authors provide controlled experiments that isolate the effect of likelihood decay, or provide a theoretical derivation linking GRPO gradient dynamics with LLD? Stronger causal evidence would significantly strengthen the paper.

2. Section 5.4 suggests that LLDS suppresses likelihood decay. Could the authors provide direct plots comparing likelihood, entropy, or gradient norms between GRPO and GRPO+LLDS to validate this mechanism?

3. In Figure 8, increasing λ does not appear to slow reward growth, and the curves are almost overlapping. How frequently is the LLDS penalty actually triggered during training? This would help clarify the practical impact of the regularization.

**Limitations:**

A key limitation of this work is that the proposed collapse mechanism remains insufficiently validated, which may affect the general applicability and reproducibility of the LLDS method. The paper attributes training collapse to Lazy Likelihood Displacement (LLD), but the current evidence mainly relies on correlation and mitigation experiments. The theoretical discussion is largely qualitative, and the paper does not provide controlled experiments or formal analysis that isolate the causal role of likelihood decay in triggering collapse. As a result, it remains unclear whether LLDS specifically addresses the underlying mechanism or simply acts as a form of generic regularization. This uncertainty raises questions about whether LLDS would consistently work across a broader range of tasks, models, and RL settings, and whether the reported results can be reliably reproduced under different training configurations.

**Strengths And Weaknesses:**

**Strengths**

1. The paper identifies a stable pattern before collapse:
   the likelihood of correct answers decreases, entropy increases, and reward eventually collapses.
   This observation itself has value and may encourage further research on the training dynamics of agent RL.

2. The paper proposes a simple and practical mitigation method.
   LLDS is easy to implement and can be integrated into existing GRPO training pipelines.
   Its gated design is also reasonable. It avoids applying overly strong regularization during normal training.

3. The experimental results are impressive.
   From the provided reward curves, the method avoids collapse at early training steps.
   The final results on General QA and Multi-Hop QA also outperform strong baseline methods such as Tree-GRPO.

**Weaknesses**

1. Lack of rigorous causal validation and theoretical analysis.\
   The paper claims that LLD is the core mechanism driving collapse. However, the evidence mainly consists of correlation and mitigation experiments. Controlled experiments are needed to isolate the effect of likelihood decay. Alternatively, the paper should provide a theoretical derivation that connects GRPO gradient dynamics with LLD.
   Sections 4 and 5 explain the LLD phenomenon and the death spiral hypothesis, which provide some intuition, but they do not include causal experiments with controlled variables. The “death spiral” dynamics are largely descriptive rather than mathematically modeled. The theoretical analysis should be more rigorous. Appendix A only provides qualitative derivations and does not prove that LLD directly leads to collapse.

2. Novelty is limited relative to prior work.\
   The paper *“Lazy Likelihood Displacement in LLMs: A Bottleneck in Tool-Integrated RL”* already introduces the concept of Lazy Likelihood Displacement (LLD) and studies likelihood decay in RL training. In this submission, the authors provide additional empirical analysis and introduce LLDS regularization.

3. Lack of direct experimental observations.\
   Section 5.4 explains that the goal of LLDS is to penalize likelihood decreases. However, the experiments do not show a dedicated plot demonstrating that correct-answer likelihood becomes stable or increases after applying LLDS. The experimental section mainly shows reward dynamics under different regularization strengths, rather than directly comparing likelihood or gradient behavior.

4. Figure 8 raises questions about the effect of LLDS.\
   In Figure 8, increasing λ does not appear to slow reward growth, and the curves almost overlap. Is this expected? One possible explanation is that LLDS is rarely triggered and therefore has only a small effect on training ?

---

> ### Author Rebuttal · Authors · 2026-03-29
>
> Thank you for your evaluation and suggestions. We truly appreciate your time and address your comments below.
>
> **Supp link** : https://anonymous.4open.science/r/ID6198_Rebuttal_Supplementary-1574/ID6198_Rebuttal_Supp.pdf
>
> ----- W1&Q1----
>
> **Answer**: We appreciate this concern and would like to clarify the causal and theoretical evidence already present in the paper, along with new results added in response to your feedback.
>
> **Controlled experiments on LLDS**:  Figures 2 and 7 should have addressed your concern on the controlled comparison: vanilla GRPO exhibits LLD followed by collapse, whereas GRPO+LLDS, identical in every other respect, stabilizes training and avoids collapse. Following your suggestion, we now plot the likelihood trajectory of correct responses directly (Supp link. Fig. 1). This confirms that LLDS stabilizes correct-response likelihood, while vanilla GRPO shows sustained decay.
>
> **Theorem A.1 links GRPO gradient dynamics to LLD**: It analyzes how the GRPO gradient affects the likelihood of correct responses, showing that their reduction is driven by two factors: high embedding similarity between correct and incorrect responses, and low likelihood of responses. Empirically, Fig. 5 supports the role of similarity-induced interference. Figs. 6 and 9(a), and lines 706–714 support the low-likelihood effect predicted by Theorem A.1.
>
> **Death Spiral as a formal definition**: Definition A.3 (App. A.2) specifies three precise conditions, persistent likelihood decay, confidence erosion under response similarity, and self-amplifying decay, that collectively define the spiral. We agree that a full closed-form theorem would be ideal; however, this would require assumptions about likelihood decay rates and similarity evolution that are difficult to justify in general.  We therefore provide a definition and support each condition with empirical evidence (Figs. 2, 3a, 4, 5, 9a). We will discuss this more explicitly in the revision.
>
> ----- W2------
>
> **Answer**:  Thank you for raising the point. We were unable to find the work you suggested, entitled “Lazy…” in any venue or preprint. As far as we know, we are the first to identify LLD as a source of collapse in tool-integrated RL and to propose a lightweight targeted fix.
>
> ----W3&Q2-----
>
> **Answer**:  Thank you for this suggestion. Following your suggestion, we plot the likelihood trajectory of correct responses. As shown in Fig. 1 in Supp. link, LLDS stabilizes the likelihood of correct responses, while vanilla GRPO exhibits severe LLD. We will include this result in the revised version.
>
> ------ W4&Q3 ------
>
> **Answer**:  Thank you for your question. **LLDS is designed to impose minimal**, selective regularization at three levels: response level, it is applied only to preserving responses with non-negative advantage; action level, it is triggered only when an action’s total likelihood decreases; and token level, once triggered, it regularizes only the tokens whose likelihood decreases, rather than all tokens uniformly. As a result, LLDS does not substantially disturb the main reward-optimization trajectory, which is why the reward curves remain close. This is a strength of the method: LLDS stabilizes training without noticeably slowing reward growth.
>
> At the same time, this does not mean LLDS is rarely triggered. Vanilla GRPO clearly collapses, while stronger LLDS regularization prevents this failure, showing that the issue is substantial rather than negligible.  We also plot the LLDS loss over training (Fig. 2  in Supp. link), and it remains consistently nonzero, indicating that **LLDS is continuously correcting likelihood-reducing token updates throughout training**.
>
> Therefore, the **overlap of the reward curves should be a merit rather than a weakness**: LLDS is active when needed, but because it is targeted rather than aggressive, it preserves the original optimization behavior while preventing collapse.
>
> ------ L1------
>
> **Answer**: Thank you for raising this point. We believe the current results already provide meaningful evidence of generalization:
> - Sizes: 3B and 7B (Tabs. 1–2).
> - Families: Qwen and Llama (Tab. 4, App. B.4.3).
> - Alignment stages: Base and Instruct variants (Tabs. 1–2, Fig. 7).
> - RL objectives: GRPO and GSPO (Tab. 1, bottom rows).
>
> While our paper focuses on agent search,  to address the concern about task generalization beyond search, we conducted new experiments on a kernel generation task (code optimization). The same collapse pattern emerges under vanilla GRPO as shown in Fig. 3  in Supp. link. Adding LLDS stabilized the training and led to performance gain as shown below:
>
> | Method            | Correctness | Fast\@1 | Fast\@1.2 | Fast\@1.5|
> |-|-|-|-|-|
> | GRPO              | 0.356   | 0.244  | 0.133 | 0.019 |
> | GRPO+LLDS  | **0.598**  | **0.452** | **0.168** | **0.084** |
>
> These results suggest that LLD-induced collapse and the effectiveness of LLDS extend beyond search-integrated QA to structurally different task domains.

---

> > ### Author Rebuttal · Reviewer_qJTK · 2026-04-04
> >
> > Thank you for your reply. The rebuttal addresses my main concerns.

---

> > > ### Author Response · Authors · 2026-04-04
> > >
> > > Thank you for your positive feedback and for recognizing our contributions. We are very happy to know that our responses have addressed your concerns.
> > >
> > > We sincerely wish you would consider adjusting your score or let us know if there are any remaining questions we could address.
> > >
> > > We also thank you for your valuable suggestions, which will help us further enhance the paper.

---

### Decision · Program_Chairs · 2026-04-30

**Decision:**

Accept (regular)

**Comment:**

This paper investigates the problem of likelihood collapse in GRPO training for tool-integrated reinforcement learning. The study identifies Lazy Likelihood Displacement (LLD), a systematic reduction in the likelihood of responses, as the core mechanism driving this failure. To mitigate this issue, the authors introduce LLD Suppression (LLDS), a fine-grained regularization that conditionally penalizes likelihood decreases to stabilize the training process.

The reviewers acknowledged that addressing training collapse in agentic reinforcement learning is a highly relevant and important problem. The empirical observations identifying the pre-collapse LLD dynamics are considered valuable. Furthermore, the proposed LLDS method is considered for being simple, practical, and highly effective, yielding significant performance improvements across multiple benchmarks.

Initial reviewer concerns primarily centered around the lack of rigorous theoretical validation linking LLD directly to the collapse, perceived limited novelty compared to prior single-turn LLD studies and the narrow focus on question-answering tasks using specific model architectures. During the rebuttal phase, the authors broadened the empirical scope by providing new experiment results on a kernel code generation task and preliminary findings on a Mixture-of-Experts architecture. They also clarified the fundamental differences introduced by multi-turn tool-integrated settings and provided direct plots of likelihood trajectories to support their mechanistic claims.

Given the strong empirical validation and the authors' satisfactory rebuttal that successfully expanded the scope of evaluation, we believe the submission represents a solid, highly practical contribution that deserves acceptance.